



# Estimating exposure of residential assets to natural hazards in Europe using open data

Dominik Paprotny[1], Heidi Kreibich[1], Oswaldo Morales-Nápoles[2], Paweł Terefenko[3], and Kai Schröter[1]

[1]Section Hydrology, Helmholtz Centre Potsdam, GFZ German Research Centre for Geosciences, Telegrafenberg, 14473 Potsdam, Germany

[2]Department of Hydraulic Engineering, Faculty of Civil Engineering and Geosciences, Delft University of Technology, Stevinweg 1, 2628CN Delft, The Netherlands

[3]Institute of Marine and Environmental Sciences, University of Szczecin, Adama Mickiewicza 18, 70-383 Szczecin, Poland

**Correspondence:** Dominik Paprotny (paprotny@gfz-potsdam.de)

**Abstract.** Natural hazards affect many types of tangible assets, the most valuable of which are often residential assets, comprising buildings and household contents. Yet, information necessary to derive exposure in terms of monetary value at the level of individual houses is often not available. This includes building type, size, quality or age. In this study, we provide a universal method for estimating exposure of residential assets using only publicly-available or open data. Using building footprints (polygons) from OpenStreetMap as a starting point, we utilized high-resolution elevation models of 30 European capitals and a set of pan-European raster dataset to construct a Bayesian Network-based model that is able to predict building height. The model was then validated with a dataset of: (1) buildings in Poland endangered by sea level rise, for which the number of floors is known, and (2) a sample of Dutch and German houses affected in the past by fluvial and pluvial floods, for which usable floor space area is known. Floor space of buildings is an important basis for approximating their economic value, including household contents. Here, we provide average national-level gross replacement costs of the stock of residential assets in 30 European countries, in nominal and real prices, covering years 2000–2017. We relied either on existing estimates of the total stock of assets or made new calculations using the Perpetual Inventory Method, which were then translated into exposure per $m^2$ of floor space using data on countries' dwelling stocks. The study shows that the resulting standardized residential exposure values provide much better coverage and consistency compared to previous studies.

## 1 Introduction

Residential assets are typically the most valuable components of national wealth (Piketty and Zucman, 2014). In Europe, dwellings contain 46 % of the gross value of tangible fixed assets (Eurostat, 2019a). Apart from dwellings, residential assets are composed of consumer durables, often referred to as household contents (Kreibich et al., 2017). These are durable goods used by households for final consumption (Eurostat, 2013). Altogether, residential buildings and their contents tend to constitute the largest share of damages induced by natural hazards. For example, 60 % of flood damages and 59 % of windstorm damages (based on the value of insurance claims) caused by hurricane Xynthia in France in 2010 were related to damages to households. This fraction is significantly larger than damages to businesses (32 % and 37 %, respectively) or automobiles (FFSA/GEMA,





2011). During the 2007 summer floods in the United Kingdom households suffered an estimated 38 % of the total value of direct
and indirect damages, while companies represented 23 % and public infrastructure with critical services 22 % (Chatterton et al.,
25  2010).

Modelling damages to residential buildings requires quantifying their exposure in terms of monetary value. This is partic-
ularly important as exposure was found to be the primary driver of long-term changes in damages due to natural hazards in
Europe and other continents (Paprotny et al., 2018b; Pielke and Downton, 2000; Weinkle et al., 2018; McAneney et al., 2019).
Exposure represents the value of assets at risk of flooding and is analysed with a variety of methods. More than half of flood
damage models identified by (Gerl et al., 2016) operated at the level of land-use classes and the remainder at the level of
individual buildings. Most commonly, also the value of assets is expressed per unit of area of a given land-use class, typically
urban fabric in context of residential buildings, obtained usually by disaggregating the stock of assets in a given country or its
subdivisions per land use units (Kleist et al., 2006; Paprotny et al., 2018a). At the level of individual residential buildings, two
distinct challenges appear: (1) obtaining building characteristics that are relevant for estimating their replacement cost and (2)
calculating the total value of a residential building and its contents.

Information on building characteristics, including floor space area, is not uniformly available. Many studies rely on national
or local administrative spatial databases such as cadastres which record multiple characteristics of buildings such as occupancy,
usable floor space or number of floors (Elmer et al., 2010; Fuchs et al., 2015; Paprotny and Terefenko, 2017; Wagenaar et al.,
2017). 3D city models can also provide the dimensions of buildings to support estimating exposure, but only in the few locations
that have such models (Schröter et al., 2018). Crowd-sourced databases such as OpenStreetMap could be an alternative, though
their utility is limited by frequently missing information on occupancy and size of buildings. Attempts have been made to
combine building footprints with other pan-European datasets such a population or land use to improve exposure estimation
(Figueiredo and Martina, 2016), but they lack scalability as they still require some locally-collected data.

Values of residential buildings are typically compiled per particular case study. A typical source of this information are local
insurance industry practices (Thieken et al., 2005; Totschnig et al., 2011). Approaches vary from assigning uniform value per
building to regression models considering building size, type and quality (Röthlisberger et al., 2018). Frequently, exposure
is computed by multiplying the building's useful floor space area by a fixed value per unit area, which in turn is taken from
national statistical institutes, government regulations, surveys of construction costs or disaggregation of the national stock of
buildings, using either gross or net values (Paprotny and Terefenko, 2017; Huizinga et al., 2017; Röthlisberger et al., 2018;
Silva et al., 2015). European-wide information on the subject is scarce. Huizinga (2007) compiled existing national estimates
of building values and filled missing data for most countries using gross domestic product (GDP) per capita. This approach
was extensively used for e.g. pan-European flood risk studies (Feyen et al., 2012; Alfieri et al., 2016) and later extended to
the whole world (Huizinga et al., 2017). Additionally, Huizinga et al. (2017) reported values of residential buildings for many
countries based on surveys by two construction companies. Ozcebe et al. (2014) also provided building replacement values for
a single reference year based on construction cost manuals and reported stock of different building types in European countries.
Finally, almost no information at all is available regarding the value of household contents. Huizinga et al. (2017) suggested,
following literature analysis, to assume that the content is worth 50 % as much as the building. For application to flood damage





modelling in Germany, Thieken et al. (2005) used household insurance reference values as a basis of estimating the value of household contents. Yet, no pan-European dataset on the topic has been created so far.

In this paper we develop a universal method of estimating exposure of residential assets at the level of individual buildings. It covers both building structure and household contents for application, at the very least, to the European Union (EU) member states. We focus on the approach that considers the total value of buildings and contents as a product of usable floor space area of a building and the average gross replacement cost of buildings and contents per m$^2$ in a given territory. Additionally, we use only publicly available datasets to achieve the task. The methodology is applicable to any location within the 30 countries covered by this study. Building size estimation routine is validated on a set of natural hazards-related case studies. Our estimates of the current gross replacement costs of building and household contents are provided at national level from 2000 to 2017 to facilitate their use in assessments of past natural disasters.

## 2   Materials and methods

This section firstly describes how residential buildings were identified using open data, then how this information is used to 70 derive the size of the buildings and finally how average values of building and household contents are obtained utilizing national accounts and demographic data. Finally, datasets and measures used to validate the building size predictions and to compare the values of residential assets with previously published estimates are described. Unless otherwise noted, all references to values of residential assets in this paper pertain to the gross stock (without loss of value due to depreciation) at current replacement costs.

### 75  2.1   Identification of residential buildings

Applying a building-level damage model requires information on the analysed objects such as size and value. Before those quantities could be calculated, residential buildings have to be identified in the area of interest. Variety of cartographic sources could be used depending on local availability, from governmental databases to topographic maps and remote sensing. The problem of accurately identifying buildings and occupancy, especially with open data, is outside the scope of this paper as 80 this issue is still subject to intense research (Schorlemmer, 2017). Here, we use OpenStreetMap (OSM), which is an openly available, crowd-sourced online database of objects constituting the natural and artificial environment of the Earth's surface (OpenStreetMap, 2019). Though created primarily by volunteers, it also contains spatial data imported from governmental GIS databases for some cities, regions or even whole countries (e.g. resulting in exceptionally comprehensive data on buildings in the Netherlands). In the context of this study the data of interests are buildings represented in a vector layer of building 85 footprints. Occupation of buildings (residential and other) is not always indicated, but can be further identified using land use information also contained in OSM. We obtained the OSM building and land use layers to develop the building size estimation method as well as the validation case studies. The download was carried out through Overpass API, a system that allows obtaining custom selections of OSM data (OpenStreetMap Wiki, 2019). Data retrieval and processing into other GIS formats was done with open-source solutions, namely Python with GDAL/OGR tools.



## 2.2 Building size estimation

Once residential buildings, i.e. their footprints are obtained, their size in terms of usable floor space area needs to be derived. The usable (also called useful) floor area of a dwelling is the total area of the rooms, kitchen, foyers, bathrooms, and all other spaces within the dwelling's outer walls. Cellars, non-habitable attics and, in multi-dwelling houses, common areas are excluded (OECD, 2019; Statistics Poland, 2019). This information is not directly available; it can be indirectly estimated from building height or the number of floors. Yet, those variables are very rarely recorded in OSM and typically not accessible from other sources either. A method of estimating building height and consequently the number of floors of a building from publicly available datasets was therefore devised here, so that the floor space area could be computed as a product of building footprint area and the number of floors. A predictive model was created by building a Bayesian Network (BN) correlating the variable of interest – building height – with seven candidate variables obtained from OSM and pan-European spatial datasets (Table 1).

A Bayesian Network is a graphical, probabilistic model which allows multivariate dependency analysis, provides uncertainty distributions of the predictions made with it. BNs are directed acyclic graphs consisting of nodes (representing random variables) and arcs indicating the dependency structure (Hanea et al., 2006). Here, we use a class of BNs known as non-parametric Bayesian Network which are quantified with empirical margins and normal (Gaussian) copulas as a dependency model. The copulas are parametrized using Spearman's (conditional) rank correlation coefficient. This class of BNs is for continuous variables only. For the purpose of this study, we use our own implementation of non-parametric Bayesian Networks as a Matlab code, the mathematics of which are described in Hanea et al. (2015).

Building height was derived from a high-resolution digital surface model "Building Height 2012" by Copernicus Land Monitoring Service (2019), which is available for 30 European cities (all European Union members' capitals plus Oslo and Reykjavik). Residential buildings (identified either through the buildings or the land use layers of OSM) for each location were extracted and a random 2 % sample was drawn to reduce to size of the dataset and ensure an even spatial distribution of samples within the city limits. The final sample has 47,466 records. Variables for the model were chosen first based on the unconditional rank correlation matrix (Table S1) and then analysing the (conditional) rank correlations between variables.

The final model is presented in Fig. 1. Building height (H) has the highest rank correlation (0.47) with population density per 1 km grid (POP). Only two variables have conditional rank correlations with building height larger than 0.1, namely building footprint area (B) and soil sealing (or imperviousness) per 100 m grid cell (IMD). Additionally, soil sealing was highly correlated with population density, but not with building footprint. The dependencies defined in the model can be explained theoretically as follows. Firstly, high population density was highly correlated with height, as one might expect the presence of tall residential buildings (high-rises, tower blocks) in densely-populated cities. High buildings also typically have a large footprint compared to single-family houses. Finally, the height of buildings is correlated with soil sealing, as urban districts with apartment blocks are largely covered by artificial surfaces proving supporting services to the buildings, such as roads, sidewalks, parking lots etc. Such surfaces reduce the perviousness of the soil. On the other hand, small single-family houses are rather found in less-densely build-up and populated suburban zones.





The accuracy of the model is analysed in section 3.1. Predicted building height was transformed into floor space area F using the following empirical formula:

$$F = 0.7 \left( \left\lfloor \frac{H - 3.3}{2.4} \right\rfloor + 1 \right) B \qquad (1)$$

where $H$ is the building height in metres and $B$ is the building footprint area (m$^2$). The $\lfloor \rfloor$ function indicates rounding down the value in brackets to the nearest integer. It is assumed that the average height of floors is 2.4 m, except the first floor which is assumed 3.3 m including floor elevation above ground, which was found to be 90 cm on average for German households affected by floods between 2002 and 2014 (see section 2.4.2). Eq. 1 further includes an allowance for the fact
that not all floor space of a building is useful, as it can contain common spaces or other non-habitable spaces. Such non-usable spaces are assumed to be 30 % of total floor space. The values of 2.4 m and 30 % were based on Figueiredo and Martina (2016), who analysed building sizes in Italy, and the comparison between observed and predicted (using methodology described herein) number of floors and usable floor space of validation case studies of flood-affected houses in Poland, Germany and the Netherlands (sections 2.4.2 and 3.1).

The described routine can be applied to any location in Europe for which any of the three explanatory variable is available. In fact, all data should be available at least for the European Union countries: building footprint from OSM or other databases and soil sealing/gridded population from pan-European datasets. An example application of the model to exposure computation is shown in section 3.3.

## 2.3 Valuation of buildings and household contents

When the floor space of a building is known, it is multiplied by the average replacement cost of dwellings and household contents per m$^2$. The total floor space of dwellings in a country is available for European countries due to recording of this information in population and housing censuses, sometimes also in household surveys (Eurostat, 2019a). This data has to be gathered from national statistical institutes, as it is not collected by Eurostat. Some countries only disseminate floor space information at census dates (e.g. Italy, Portugal, Spain), while others from surveys carried out less frequently than annually
(e.g. France) or only as part of the EU Survey of Income and Living Conditions (e.g. Norway, Sweden). There are also countries that calculate continuous balances of housing stock or extract data from housing registers, thus providing annual time series of floor space area in the country (e.g. Denmark, Germany, the Netherlands, Poland, Romania). Finally, for some countries only household floor space data from the 2012 edition of the EU Survey of Income and Living Conditions were available (e.g. Belgium, Norway, Sweden). Information on the data collected on dwelling stock is provided in Table S2.

### 2.3.1 Residential buildings

Statistical institutes in most European countries are recording the stock of fixed assets, including dwellings, for purposes of national accounting (Eurostat, 2013). Annual time series of the gross stock of dwellings is available for 22 EU countries from Eurostat, though the data for two countries – Latvia and Poland – couldn't be used due to major methodological differences





which are discussed in Table S3. The value of dwellings is provided from the aforementioned resource in nominal and previous

year's prices. A deflator to obtain real (2015) prices was constructed based on the two timeseries. Finally, the value of all dwellings was divided by the total floor space area in a country to obtain average value per m$^2$. The method does not consider building types or quality, but this information is scarcely available from open datasets on buildings. Information on specific data sources on dwelling values is provided in Table S2.

The remaining EU countries and three other Western European nations (Iceland, Norway, Switzerland) required more data

collection efforts. According to the European System of Accounts (ESA) 2010 manual (Eurostat, 2013), the Perpetual Inventory Method (PIM) should be applied whenever direct information on the stock of fixed assets is missing. In practice, most countries use PIM to arrive at the stock estimates that are published through Eurostat (Eurostat and OECD, 2014). PIM accumulates past investments over time to indirectly estimate the value of the stock (U.S. Department of Commerce. Bureau of Economic Analysis, 2003). The general formula for PIM to obtain the gross stock is as follows (National Bank of Belgium, 2014):

$$S_t = \sum_{j=0}^{L}(I_{t-j}G_j) \qquad (2)$$

where:

$S$ denotes stock of an asset;

$t$ is the calendar year;

$j$ is an annual increment;

$I$ is investment in year $t-j$;

$L$ is the maximum service life of an asset in years;

$G$ is the proportion of an asset purchased in $t-j$ and still in use in $t$.

Three quantities are needed to obtain the stock of dwellings $S$: investment in housing, an estimate of the dwellings' service life and the fraction of dwellings of the same vintage that are retired every year. Investment (gross fixed capital formation for

asset type 'dwellings') is available from Eurostat, national statistical institutes or country-specific research estimates. However, sufficiently long investment series were only identified for Sweden, while for other countries had to be extrapolated using total investment or gross domestic product (GDP), a method which is also applied by national statistical institutes when necessary (Eurostat and OECD, 2014; Rudolf and Zurlinden, 2009).

Parameters $L$ and $G$ are assumptions that usually stem from estimates of average service life of assets. Most national

statistical institutes derive $G$ by assuming certain probability distributions known as retirement patterns or survival functions. This means that a different proportion of dwellings is retired each year, with the highest proportion around the average service life. However, this requires assuming a certain probability distribution, and national methodologies indicate a large variety of those (normal, lognormal, Gamma, Weibull, Winfrey etc.). Further assumptions have to be made regarding the distribution's dispersion and maximum service life (OECD, 2009). It also vastly increases the length of investment time series necessary to

apply PIM, which would require collecting investment series going back even to the early 19th century. In effect, some countries





with short data series apply no survival function (Eurostat and OECD, 2014). This approach is known as "simultaneous exit" and assumes that all assets are only retired when reaching a given service life. Eq. 2 is therefore simplified to:

$$S_t = \sum_{j=0}^{L_{mean}} I_{t-j} \qquad (3)$$

which now only requires assuming an average service life of dwellings $L_{mean}$. As a sensitivity check, we applied a lognor-
mally and normally-distributed retirement pattern to the Swedish investment series, the longest we have collected. We assumed
a dispersion factor from 2 to 4 (i.e. ratio of mean and standard deviation of service life) and maximum service life equal to
twice of the average, as suggested by the National Bank of Belgium (2014). The calculation yielded a gross stock of dwellings
in Sweden in 2017 lower by 5–15 % compared to an estimate derived with no survival function. Consequently, we relied on the
simplified method to apply PIM for six countries (Iceland, Malta, Norway, Spain, Sweden and Switzerland). $L_{mean}$ for each
country was taken from national methodologies collected in a survey by Eurostat and OECD (2014), except for Switzerland,
which was taken from Bundesamt für Statistik (2006).

For further four countries, where data on investment is limited, but the balances of the number of buildings and their floor
space is available, a modified PIM was applied. In those cases, we computed an initial estimate of the stock of dwellings
(Bulgaria in 1999, Latvia in 2000, Poland and Romania in 1995) based on national construction costs in the base year and then
used annual data on investments in, and retirement of, dwellings in the country to arrive to a timeseries of the gross stock. In
this case eq. 2 becomes:

$$S_t = S_{t-1}(1 - G_t) + I_t \qquad (4)$$

here $G_t$ is the fraction of the stock retired during year $t$. In this way, service life assumption and long data series are not
needed, with the drawback of assuming uniformity of the existing stock of dwellings and that all investment goes into building
new dwellings rather than also into renovation of dwellings. We also tested the method from eq. 4 using extrapolated investment
series, but it yielded far lower estimates of building asset values which were also much lower than for neighbouring Central
European countries. With a modified PIM, the exposure estimates were more closely aligned to countries at a similar level of
development. Calculation for the remaining country, Croatia, was not possible due to the lack of even basic data needed for the
computation. Data sources and assumptions for individual countries are provided in Table S2.

### 2.3.2 Household contents

Data availability for the stock of household contents is much lower than for dwellings. This item is termed in national ac-
counting as 'consumer durables' and assumed to be consumed within the accounting period, rather than accumulated, as those
durables are not relevant from the perspective of economic production processes. As such, they are considered memorandum
items in ESA 2010 (Eurostat, 2013) and consequently few European countries have published national estimates of the stock of





consumer durables, namely Estonia, Germany, Italy and the Netherlands (OECD, 2019). Yet, even those few available datasets include personal vehicles in the stock. Cars and motorcycles are typically located outside the residential buildings, hence including them in estimates disaggregated by m$^2$ of floor space would not be suitable. Further, they are insured separately from houses and their contents, therefore not included e.g. in reported flood damages from post-disaster household surveys (Thieken et al., 2005; Carisi et al., 2018; Wagenaar et al., 2018). Given all those constraints, we calculate our own estimates of the stock

of household contents (durables) for all countries included in the study.

In order to estimate the stock of household contents, the PIM method is applied again. However, the contents consist of various durables of different service lives, therefore eq. 3 has to be rewritten as:

$$S_t = \sum_{a=1}^{A} \sum_{j=0}^{L_a} I_{t,a-j} \tag{5}$$

where the stock of household contents equals the sum of stocks for items $a = (1, ..., A)$, each with service life $L_a$. No

retirement pattern was assumed, hence all items are included in the stock until reaching their average service life. The data on annual investment was gathered from final consumption expenditure of households split according to the Classification of Individual Consumption by Purpose (COICOP). The relevant durables are a set of twelve items at COICOP 4-digit level, i.e. all durables less items under code 07.1 "Purchase of vehicles". However, only Sweden publishes annual data with such level of detail; data disaggregated at COICOP 3-digit level are disseminated for 28 countries, at COICOP 2-digit level for Switzerland

and no data are available for Croatia. We therefore computed the average share of spending on durables within COICOP 3-digit categories using 5-yearly household survey data from Eurostat on detailed consumption expenditure patterns per country. The same approach was previously applied by Jalava and Kavonius (2009) to estimate the stock of durables in Europe. It allowed us to estimate spending on durables from COICOP 3-digit data. Assumptions about service life of durable items (aggregated to COICOP 3-digit items) were calculated from German estimates presented by Schmalwasser et al. (2011). We averaged

1991 and 2009 estimates of service lives from that study and weighted the COICOP 4-digit items according to their share in spending. The service life of appliances for personal care (COICOP code 12.1.2) was not provided in the aforementioned resource, hence it was taken from Jalava and Kavonius (2009). A list of durable items, assumptions on their service life and the share of spending on durables per COICOP 3-digit items are shown in Tables S4 and S5. For Iceland detailed consumption expenditure surveys are not available, hence average share in 15 EU members states was used instead.

Final consumption expenditure data were collected from Eurostat, OECD and national statistical institutes. Due to the very long estimated service life of durables in the 'personal effects' (COICOP code 12.3.1) category (45 years), the spending on those items had to be extrapolated using data on total private consumption expenditure, or GDP. This should have, however, limited influence on the results for recent years given the rather small share of spending on durable personal effects. For France, which has detailed expenditure going back to 1959, truncating the data to 1995 (the minimum availability for the countries

considered except Malta) and extrapolating it with total private consumption resulted in a 2–5 % lower estimate of the stock of household contents, depending on the year. The uncertainty increases when moving back in time. Detailed sources of data are shown in Table S6. The calculation in eq. 5 was carried out with expenditure time series in real (2015) prices, and then





converted to nominal prices using country- and item-specific deflators. Additionally, country-specific deflators of household contents were devised from the time series of the stock of consumer durables in real and nominal prices. Those deflators can be

used to estimate the value of damages to household contents in real prices. Lastly, the stock of consumer durables was divided by the total floor space area in a country to obtain average value per m², as for residential buildings. However, for several countries, due to a large number of unoccupied dwellings (as identified in data from Eurostat (2019a)), only the floor space area of occupied dwellings or the number of households was used in this calculation. Instances of using different floor space area estimates to obtain average building and contents values are indicated in Table S2.

## 2.4 Validation of the exposure model

### 2.4.1 Validation measures

Predictions of building height, number of floors, and floor space area are compared with observations using several error metrics (Moriasi et al., 2007; Wagenaar et al., 2018):

- Pearson's coefficient of determination ($R^2$) was used to measure the degree of collinearity between predicted and ob-
served values, with higher $R^2$ indicating stronger correlation.

- Mean absolute error (MAE) was used to measure the average absolute difference between predicted and observed values, with higher MAE indicating higher error.

- Mean bias error (MBE) was used to measure the average difference between predicted and observed values, with positive MBE indicating overprediction and negative MBE indicating underprediction.

- Symmetric mean absolute percentage error (SMAPE) normalizes MAE by considering the absolute values of predictions and observations, with value close to 0 indicating small error compared to the variability of the phenomena in question.

Equations for the listed measures are shown in Table S7. For validation purposes, we use the predictions as mean (expected) values of the uncertainty distribution of the variables of interest per each data point (building). We also analyse the uncertainty of the height prediction model and perform an out-of-sample validation.

### 2.4.2 Datasets for validation and comparison

The out-of-sample validation of building heights was done individually for the 30 capital cities contained in the sample quantifying the BN. Then, a collective out-of-sample validation was done using a new 1 % sample of the buildings in those cities, which does not overlap with the sample used to quantify the BN. Predictions of building heights transformed into number of floors were validated using a large (N = 62,580) sample of residential buildings that were identified as potentially endangered

by coastal floods and sea level rise in Poland according to a study by Paprotny and Terefenko (2017). The dataset contains building polygons with the number of floors and constitutes part of the Topographical Objects Database maintained by the office of surveyor-general in Poland. It was created through combination of remote sensing, field surveys and administrative





registers and is accurate as of year 2013. The quality of the data should correspond to a 1:10,000 scale map and the quantitative

information contained in the dataset should nominally deviate from real values by no more than 20 %. For each building, the

footprint area, population and soil sealing were derived to run the BN-based model and converted into number of floors using

eq. 1.

Validation of floor space area predictions was carried out using results of post-disaster household surveys covering six

river floods and three flash floods that affected Germany between 2002 and 2014 and a river flood along river Meuse in the

Netherlands in 1993 (Thieken et al., 2005, 2017; Rözer et al., 2016; Spekkers et al., 2017; Wagenaar et al., 2017, 2018). In

the German surveys, conducted mostly in the south and east of the country, respondents were asked to provide information on

the floor spaces of their households. The floor space area of multi-family buildings was extrapolated using the total number of

flats in the building multiplied by the floor space of the surveyed household. In the Dutch survey, the information on the floor

space area was taken from the national cadastre. For each survey data point an OSM building polygon was downloaded and

other statistics necessary to run the BN model were extracted. However, both survey datasets include considerable uncertainty

related to the location of individual buildings. Therefore, the analysis was done only for buildings for which there was good

confidence that corresponding OpenStreetMap buildings were correctly identified, based on the building footprint area recorded

in the survey datasets. Also, the analysis for Dutch data was done only for single-family houses, as the floor space data for

apartment buildings only referred to particular households, not the whole buildings. As this was also occasionally the case in

the German sample, instances of floor space being less than half of building footprint were excluded. This threshold also helps

excluding residential buildings with large non-residential parts (e.g. agricultural or commercial), as was done by Fuchs et al.

(2015).

Estimates of building and contents value cannot be directly validated due to the lack of information on this subject at the

level of individual objects. We can only compare our results with other published results, which is done in section 4.1. Those

published results include two pan-European studies: (1) a flood risk assessment for the European Commission – Joint Research

Centre (JRC) by Huizinga et al. (2017) and (2) seismic risk assessment for the "Network of European Research Infrastructures

for Earthquake Risk Assessment and Mitigation" (NERA) project by Ozcebe et al. (2014). Both studies used construction

cost surveys and manuals as well as regression analyses with socio-economic factors. Additionally, we compare estimates

calculated in this study with values used in available local or national risk assessments.

## 3  Results

### 3.1  Validation of building height and floor space predictions

The exposure estimation procedure was first validated by comparing observed and modelled residential building height. This

analysis was done through an out-of-sample validation in which we use the original sample of 2 % of buildings in 30 European

capitals (section 2.2) to predict another sample of 1 % of the buildings, with no overlaps between the two datasets. Figure

2 displays a comparison between observed and modelled heights. The coefficient of determination ($R^2$) is a moderate 0.36.

310  Still, the model predicts correctly the average height (9.65 m versus 9.57 m observed), but underestimates the variation, as the





modelled sample has a standard deviation of 3.30 m versus 5.89 m found in observations. In effect, despite the negligible bias of the model in general, height of tall buildings (more than 20 m high) is mostly underestimated. Mean absolute error is 3.24 m, which is 34 % of the mean height (Table 2).

An out-of-sample was also carried out for each city in the dataset, where only the capital in question was excluded from the data quantifying the dependency structure of the BN model. The full results are provided in Table S8. Lowest $R^2$ were computed for Nicosia (0.02) and Reykjavik (0.04), though the latter has lowest average building height among the cities considered here. On the other end of the scale are Vienna (0.55) and Berlin (0.50). Relatively low errors and bias was found for e.g. Amsterdam, Luxembourg, Vienna, Warsaw and Zagreb. The largest MAE and negative MBE was identified for Bratislava (7.28 m and -5.38 m, respectively), while the highest MBE was recorded for London (+2.69 m).

The second step in obtaining floor space – the number of floors – was tested against a large number of Polish residential buildings located in the coastal zone. Results in Table 3 show that average error is slightly less than a third of the average number of floors. $R^2$ for particular building types is low, but better overall, as the method has clearly ability to distinguish small single-family house from multi-family buildings. Overall, 47.1 % of the buildings had the number of floors predicted correctly (Table 3). The number of floor is rather underestimated than overestimated, especially for higher buildings. The error does not exceed one floor for 4–6 floor buildings in around 70 % of cases. For buildings of 7 floors or more, underestimation is mostly by two floors.

Finally, predictions of the floor space area were tested against Dutch and German households (Table 3). The average error was equal to about a third of average building height of the Dutch buildings. For the German buildings, average error was almost half the average height. The size of Dutch buildings is on average slightly underestimated (-11 %), but the opposite happens for German houses (+15 %). Nonetheless, the model clearly can distinguish between single-family (detached) and multi-family houses. Larger variation in heights of apartment buildings also results in higher $R^2$ and lower SMAPE compared to detached houses which are typically quite similar in the number of floors. Mean absolute error (MAE) is still larger for multi-family houses, but bias is lower than for the other two types of buildings in the German sample.

### 3.2 Pan-European estimates of building and household contents value

As described in section 2.3, statistical data on buildings and household expenditure were collected for a study area of 30 countries (Iceland, Norway, Switzerland and the European Union less Croatia). The dataset reveals a considerable stock of residential assets in place. Based on those statistical data alone, we estimate that there were 259 million dwellings in the study area at the end of 2017, some 12 % of which are vacant or occupied seasonally. Those dwellings had a collective useful floor space area of almost 24 billion $m^2$ and were worth 36.7 trillion euro in gross replacement costs. At country level, the value of assets per $m^2$ of floor space varies substantially (Fig. 3 and 4). Iceland had the highest estimated value of dwellings per $m^2$ (2284 euro), followed closely by Germany and Finland. Differences in dwelling sizes, vacancy rates and average number of persons per household result in average home replacement costs varying even more. Icelandic dwellings, typically larger than European average, are the most expensive in Europe, though in per capita terms costs are higher in Denmark (Fig. 3a). On the other side of the spectrum, Romanian dwellings are the smallest (in terms of average floor space) and cheapest to reconstruct





(412 euro per m$^2$). Higher values are recorded in Bulgaria, Lithuania and other Central European states. Southern European and Benelux nations fall in the middle of the distribution (Fig. 4a). The stock of dwelling and their prices have grown rapidly since year 2000 (Fig. 5a). Almost 5 billion m$^2$ of floor space was added and the average dwelling size has increased as well. In nominal prices, the average replacement cost of residential buildings per m$^2$ of floor space has grown at least by 14 % (Greece) and as much as six-fold in Romania; the growth in average European dwellings costs was 53 %. In constant prices, the average

replacement cost of the existing dwelling stock has declined in four countries (Denmark, France, Luxembourg, Slovenia). The highest growth of 79 % was recorded in Slovakia. Average European replacement costs per m$^2$ has gone up by a modest 7 %. Changes of dwelling value in constant prices should be interpreted as change in the characteristics of the stock of residential buildings: its average quality, material, size and type (single- and multi-family houses, dwellings for permanent or seasonal use, etc.). There appears to be no clear pattern of the distribution of those changes, but southern countries had rather lower

rates of cost growth than the northern states. The country with the highest replacement costs per m$^2$ changed multiple times in the 17-year timeframe, alternating between Germany, Ireland, Sweden, Switzerland and finally Iceland.

Household contents in Europe is diversified collection of durable items, which we estimated were worth 6.6 trillion euro at the end of 2017. Furniture, furnishings and floor coverings constituted 39 % of the gross stock of household contents, followed by jewellery, clocks and watches (25 %); audio-visual, photographic and information processing equipment (11 %); major

household appliances (10 %) and various other tools, equipment and appliances (16 %). Variation between countries is higher than for dwellings (Fig. 4b), albeit mainly due to exceptionally large stock of consumer durables in Switzerland (666 euro per m$^2$ as of 2017). Nordic countries are less prominently featured in the top of the ranking compared to building values (Fig. 5b). The highest values are recorded, apart from Switzerland, in Austria, the United Kingdom, Norway and Germany. Switzerland also comes first in the value of contents per household and per person. Lowest stocks of durables per m$^2$ were estimated for

Hungary (84 euro), Bulgaria and Cyprus, though the latter's value is a result of large sizes of dwellings, hence an average Cypriot household has more assets than homes in other Central European countries. In nominal terms, the growth in household contents was smaller than for dwellings in nominal terms: 30 % for the growth in average European value per m$^2$, varying from decline in Ireland to an almost four-fold increase in Slovakia and Romania. Yet, many household items have seen their prices grow slowly or decline, especially for electronic equipment. In effect, an average household in Europe had 19 % more

consumer durables per m$^2$ in 2017 than in 2000, even if growth was lowered by the increase in average floor space available to households. Three countries (Italy, Luxembourg, Spain) recorded a decline (Fig. 5b), while more than tripling of contents value was recorded in Latvia and Slovakia. Growth was clearly higher in Northern and Central Europe than in Southern Europe, as consumer spending on durables is very sensitive to the countries' economic performance. Switzerland had the highest values of contents per m$^2$ throughout 2000–2017, while the lowest values were first estimated for Latvia, later Bulgaria and finally

Hungary.

### 3.3 Example application

To illustrate an application of the two components of the study – building-level height predictions and country-level valuations of residential assets – we downloaded current OSM building data for Szczecin, Poland. A city of slightly more than 400,000





people is endangered in its low-lying parts by floods and sea level rise (Paprotny and Terefenko, 2017). OSM data indicated
27,971 buildings within the city limits. After calculating the footprint area of each building, corresponding population density
and gridded soil sealing at 100 m resolution was extracted from pan-European datasets, as in section 2.2. The BN model
predicted building height for each building, which was then transformed into number of floors and consequently useful floor
space area (eq. 1). The average building was found to have a floor space of 467 m$^2$ (uncertainty range 453–482 m$^2$). The number
of residential buildings and their average size slightly larger than the values for 2017 recorded in the national statistics – 27,068
and 419 m$^2$, respectively (Statistics Poland, 2019). The floor space of each building was multiplied by the average replacement
costs of buildings and household contents in Poland in 2017, which is 683 and 109 euro per m$^2$, respectively (Tables S2 and
S6 in Supplementary Information 2). The total value of residential assets per building in a fragment of the city is presented in
Fig. 6.

Combining our exposure estimates with flood maps for extreme sea levels (Paprotny and Terefenko, 2017), we can identify
209 residential buildings in the city that exist within the 100-year flood hazard zone. Their aggregate value amounts to 19.3
million euro. Then, a flood vulnerability model can be applied to estimate damages in case of the event, e.g. pan-European JRC
depth-damage function for residential assets (Huizinga, 2007). This vulnerability model applied to water depths computed by
Paprotny and Terefenko (2017) produces an estimate of damages from a 100-year flood event amounting to 6.1 million euro.

## 4   Discussion

### 4.1   Comparison with alternative estimates of residential assets value

Estimates of residential building replacement cost per m$^2$ from two external sources, by the JRC (Huizinga et al., 2017) and
NERA project (Ozcebe et al., 2014) are gathered in Table S9 and compared with our estimates in Fig. 7. In a few cases, two
different estimates are provided by the JRC, as two construction surveys were used as source of information. Many of the JRC
dwelling values for year 2010 are similar to our calculation for the same year. In most cases, JRC provides higher estimates,
which can be the result of using information on the construction costs of modern dwellings, rather than the replacement value
of actually existing stock of housing. It is noticeable that the two alternative estimates by JRC differ substantially between
themselves, especially for Germany and Poland, with our calculation falling in the middle in those two divergent cases. NERA
project estimates (for year 2011) show much less variation between countries and almost uniformly show lower replacement
costs for Western European dwellings and higher for Central European houses. This is a result of using a set of "reference"
countries and a regression based on GDP per capita. The latter was developed for global application, in effect compressing the
variation in building costs: they vary only by a factor of 2 in the NERA estimates, despite GDP per capita in the countries in
question changing by a factor of 15 as of year 2011.

Household contents was not directly estimated by JRC in the study by Huizinga et al. (2017), but rather recommended taking
half of the dwelling value. We therefore take 50 % of JRC's building value estimates for comparison with our estimates (Fig.
7). In all cases, the resulting household contents value are much higher than our estimates. This could be partially a result of
including more items in the contents, e.g. vehicles and semi-durables, though the extent of the term was not stated in the cited



study. In our study, a ratio close to 2:1 for buildings and contents was found only in Malta (1.86), while for other countries it is at least 3:1. The average ratio is almost 5.7.

Some other literature estimated could be compared with our results. Studies based on German post-disaster surveys computed exposure based on a insurance sector guideline for residential building values deflated to a particular year with the construction price index (Thieken et al., 2005). Household contents was computed using a regression analysis of average insurance sums and local purchasing power. Average replacement costs of buildings affected by riverine floods between 2002 and 2013 was, on average, 2594 euro per m$^2$ in 2013 prices. The corresponding value for household contents was 545 euro, a ratio of 4.76:1. A weighted average of our estimates would be 1944 and 377 euro at price level of 2013. While both estimates are lower, the ratio of 5.16:1 is close to the value used in the German surveys. In a study of coastal floods and sea level rise in Poland (Paprotny and Terefenko, 2017), the authors used the average construction costs of new multi-family dwellings from the national statistical institute. Household contents was estimated on the basis of average share of consumer durables in GDP identified for some developed countries by (Piketty and Zucman, 2014) and total floor space of dwellings in the country. Their estimates of 936 and 147 euro for year 2011 are higher than 717 and 94 euro for dwellings and contents per m$^2$, respectively, computed in this study. However, the first value is based on new dwellings rather than replacement costs of existing stock, while the second value includes the cost of personal vehicles, which would add about half to our estimate of household contents (see section 4.2), thus matching the other calculation. Silva et al. (2015) used residential building replacement costs from a governmental decree, updated annually, for their seismic risk assessment. As of 2013, the values per m$^2$ separately for major cities, other urban areas and rural areas were 793, 693 and 628 euro, respectively. Given the distribution of population by regional typology (Eurostat, 2019a), that amounts to around 700 euro on average of the country, only slightly more than 671 euro calculated here.

### 4.2 Uncertainties and limitations

Predictions of floor space area involve several uncertainties along the chain of computations. Firstly, the Bayesian Network for predicting buildings was quantified based on a set of capital cities. Those cities vary enormously in size, cover 30 countries and include at least to some extent the surrounding metropolitan area, but they don't include area of more rural character. Incorporation of those areas could improve predictions for single-family houses. Also, the source elevation model has a resolution of 10 m, therefore the height of buildings with small footprint areas could be less accurately assigned to OpenStreetMap polygons. In the second step, obtaining the number of floors, a constant height of each floor was assumed, though they tend to vary to some degree (Figueiredo and Martina, 2016). Also, a more diversified set of evidence could the improve the calculation, similarly for the last step of deriving useful floor space, which depends on the assumption what percentage of the area of a building is actually used for living purposes. This is particularly problematic with building of mixed use, as first floors of residential buildings are often utilized by shops and other services.

Uncertainties related to economic valuations are largely methodological or related to limitations in availability of some data for certain countries. Most of the gross stocks of dwellings are taken directly from national estimates, which are computed with a variety of assumptions related to service life and retirement patterns as well as investment data availability, coverage





and detail. As noted in Table S4, analysis of methods identified timeseries for two countries incomparable with others, but more datasets could be affected by local methodological specifics. The stock of household contents was computed with a uniform approach, but service life assumptions based on a German study might not be suitable for other countries. Also, the availability of historical data on consumption expenditure varies between countries and most detailed COICOP 4-digit data is

not accessible on per annum basis, necessitating assumptions about share of durable spending in more aggregated data. Quality of the expenditure data could also be questioned given the very large differences between deflators for individual durable items between countries. This is most strongly visible in the data for Ireland, where prices of all items have dropped significantly since year 2000 according to national statistics, which not in line with experience of other European economies. Consequently, the estimate of the stock of household contents for Ireland is likely too low and the strong upward trend overestimated. Further,

availability of dwelling and household numbers and especially the floor space statistics is not uniform. For some countries, data on temporal changes in average floor space per dwelling or the total area are not published. Yet, housing statistics are typically better for Central European countries than Western European states, quite the opposite to economic data availability. This is likely a result of poorer living conditions in the new EU member states prioritizing gathering information on the subject compared to Western Europe, while their less-developed statistical systems usually generate lower detail and shorter time series

of economic statistics.

The study presented only valuations of dwellings and household contents as gross stock, i.e. replacement cost without allowing for depreciation of assets. Merz et al. (2010) argued that for analysing damages to natural hazards net costs should be used instead, as the value actually lost is the remaining, depreciated value of assets. This is sensible in the perspective of national accounting, where changes to net stocks of assets are of main interest e.g. for calculating GDP using the income approach or

indicators such as net disposable income of households or net savings. Still, an asset typically cannot be restored to a particular depreciated state, therefore from the perspective of those who would need to pay for repair or replacement of the damaged or destroyed assets the gross stock is a better indicator of the possible cost of post-disaster recovery. Depreciation of residential buildings varies to a large degree in Europe, not least due to very different assumptions on the patterns of depreciation. One method is called "straight-line", as it assumes an asset loses a given percentage of its gross value each year and also requires

defining a retirement pattern as in the computation of gross stock. It is the default method in the ESA 2010 system and used e.g. in Belgium, France, Germany, Italy, Portugal and the United Kingdom (Eurostat, 2013; Eurostat and OECD, 2014). The other method, "geometric", assumes that an asset loses a given percentage of its remaining (net) value and is used in e.g. in Austria, Estonia, Norway and Sweden (Eurostat and OECD, 2014). The total stock of dwellings for the 22 countries available from Eurostat's database indicates a depreciation of 37 %, varying from 22 % in France to 55 % in Hungary (Eurostat, 2019a).

Consumer durables less personal vehicles are used here for household contents on the basis of what items are actually insured and compensated after natural hazards events. Overall damages to households could be yet higher. In the aftermath of the 2010 Xynthia storm, 8 % of flood-related insurance claims were related to cars on top of 5 % of windstorm-related claims (FFSA/GEMA, 2011). In the study area, annual consumer spending on purchase of vehicles amounts to some 300 billion euro per year, 92 % of which is on motor cars (Eurostat, 2019a), hence assuming 11–12 years of service life (Schmalwasser

et al., 2011) the stock of vehicles owned by households would amount to about half of the value of other consumer durables.





Households also stock semi-durables and perishables. They are generally excluded from any assessments on household wealth due to the limited information on the usage time of items in question and their rather low value (Goldsmith, 1985). Spending on semi-durables (e.g. clothing, footwear, books, toys, small appliances) in the study exceeded 750 billion euro in 2017, therefore it would add about 10 % to the estimated stock of durables for each year of assumed service life. Spending on perishables (e.g.

food, fuel, medicines, newspapers) amounts to 2.5 trillion euro per annum in the countries considered here, but if households hold only a weekly stock of perishables, their total value is less than 1 % of the stock of durables.

### 4.3   Future outlook

Improving building height predictions for the purpose of exposure estimation would involve incorporating new sources infor-mation. For building heights, lidar scanning results from smaller cities and rural areas should be incorporated to increase the

diversity of the sample for a Bayesian Network model. The model itself could also be built separately based on data of different typology (urban, suburban, rural) or for different parts of Europe. More diversified resources are needed as well to analyse the relationship between building height and the number of floors and further with the usable floor space of the building, which can differ between countries and building types. As a more immediate step, the code used in this study is expected to become publicly available to facilitate its application and further testing.

Time series of building and contents value provided in this study (Supplementary Information 2) have several applications. The main use is providing economic valuation economic assets for natural hazard exposure and risk assessments carried out at the level of individual buildings (large-scale mapping). The time series could be used to correct past recorded damages from natural disasters both for changes in asset reconstruction costs (separately for dwellings and contents) over time, but also changes in average quality of residential buildings and incomes of households that translate into more expensive consumer

durables kept at home. Finally, the data could be used to rescale absolute damage functions, which generate damage estimates based on intensity of the hazardous event not as percentage of assets lost, but as an absolute value for a given country in a specific year. In the field of flood risk, almost half of damage functions provide absolute values of damages instead of relative values (Gerl et al., 2016). With our data, for instance, flood damage curves for the United Kingdom at price levels of 2012 (Penning-Rowsell, 2013) could be applied to a German flood in 2002 by using the ratio of average replacement costs of

residential assets in the UK and Germany in the respective years and currencies.

Further research on countries with good economic data would involve expanding the coverage in multiple aspects. Themat-ically, net (depreciated) value of residential assets could be added to the dataset, as most of the necessary data have already been collected here. Net stock of dwellings is directly available for four more countries than the gross stock (Norway, Spain, Sweden and Switzerland), while for others PIM method would be used (Eurostat, 2013; Eurostat and OECD, 2014). Net stock

of consumer durables can be computed from the same data as gross stock (Jalava and Kavonius, 2009). Estimates of the stock of vehicles (gross and net) could be added, disaggregated on per household or per capita basis. Spatially, some developed non-European countries could be added which disseminate necessary data e.g. through OECD. It should be possible to add Croatia and EU candidate countries to the dataset once they start publishing detailed EU-mandated national accounts data. In the temporal dimension, the dataset could be extended in the past at least for certain countries with long data series (partic-





ularly France and the Nordic countries), so that it could be applicable natural hazard case studies that have occurred before
year 2000. An analysis of the trends in the stock of residential assets and economic factors determining it could possibly also
provide insights how could it change in the future, for the benefit of projections of natural hazard risk under climate change.
Furthermore, the possibility of regionalization of asset values could be investigated. Some countries have disseminated building
stock data at regional level (e.g. Germany, Poland, Spain), which could lead to analysis what regional economic data could be

used to predict inter-country variations, e.g. GDP, gross fixed capital formation, compensation of employees in the construction
sector, disposable income of households etc. Combined with gridded population and land-use datasets, these estimates could
result in detailed dasymetric exposure mapping in Europe (Kleist et al., 2006; Thieken et al., 2006). Finally, incorporation of
more detailed building characteristics, where available, could be applied to differentiate estimates of building value per m$^2$.
Some countries distribute investment or stock data split by various types of buildings, which can be incorporated also in the

PIM method. Several countries use different service life assumptions according to building type (Czechia, Estonia, Sweden),
ownership (Latvia, Slovenia), age (Denmark, Germany) or material (some non-European countries), according to a survey by
Eurostat and OECD (2014).

Still, most countries of the world do not disseminate so detailed housing, asset, investment or expenditure data as were used
in this study. Simplified methods to indirectly estimate exposure will therefore be needed. GDP per capita was incorporated by

the NERA study as such measure, but as the comparison in section 4.1 has shown, this is not necessarily a good indicator. Also,
there are significant variations between value of residential assets per m$^2$ compared to GDP per capita and further differences
between the composition of those assets. They vary by a factor of 4 and 6, respectively. The lowest exposure relative to GDP
per capita is recorded particularly in countries where GDP is far higher than actual income of their population, like Ireland
and Luxemburg. Using e.g. final consumption expenditure of households per capita as a proxy gives better results, reducing

the variation in total residential assets per m$^2$ between countries to a factor of 2.6. Estimates of this variable are available
globally e.g. from the National Accounts Main Aggregates Database (United Nations, 2018). More detailed data is available
approximately each five years from the International Comparison Programme, including household expenditure at COICOP
2-digit level. Developing such simplified approaches requires further analyses. Until then, we can propose the following rule
of thumb based on the average asset values in the European countries, that the total residential assets per m$^2$ equal 6 % of GDP

per capita, of which one-sixth are household contents.

## 5   Conclusions

In this study we have explored aspects related to estimating exposure of residential assets in Europe. Firstly, we proposed a
methodology to estimate useful floor space area of buildings in situation when the only accessible quantitative measure about a
house is its footprint area. This basic measure can be derived from various sources, from analogue topographic maps to crowd-

sourced databases like OpenStreetMap (OSM). Building height or the number floors is only occasionally accessible, hence it
has to be estimated based on other information. In our work, we have shown that a Bayesian Network quantified with a set
of publicly available pan-European raster datasets and building footprints from OSM has the ability to differentiate between



urban high-rises and suburban or rural single-family dwellings. Further, it can be applied to approximate building dimensions that can be the basis for assigning economic value to assets in question.

In the second part of the analysis, we harnessed publicly disseminated statistical data on housing stock and national economies to calculate time series of average value of residential assets—building structure and household contents—for 30 European countries. It can be applied whenever local exposure data are missing or no detailed characteristics of buildings are accessible. Additionally, it can improve analyses of past natural disasters by estimating exposure of assets in a particular year and country, as well as enable transferability of damage models that provide absolute rather than relative damages. More

work is expected on expanding the thematic, spatial and temporal coverage and resolution of the dataset. It will be also applied as an important basis for constructing and validating a new generation of vulnerability models in natural hazards.

*Code and data availability.* This study relied entirely on publicly-available datasets, with the exception of validation datasets from Poland, Germany and the Netherlands in section 3.1 (see Acknowledgments). Data sources for the building height model are provided in Table 1. Detailed data sources for the economic computations are listed per country and variable in Supplementary Information 1. The full estimates

of residential asset values are provided in Tables S1–S8 in Supplementary Information 2. Uninet software used to analyse and visualize the BN model is available from LightTwist Software for free for academic purposes (http://www.lighttwist.net/wp/). Implementation of the BN in Matlab is available from the authors upon request until it becomes publicly available.

*Author contributions.* DP conceived and designed the study, collected and analysed the data and wrote the first draft of the manuscript. HK and KS helped guide the research through technical discussions. OMN provided code for data analysis and was involved in technical

discussions. PT provided, and supported processing of, some of the spatial datasets. All authors reviewed the draft manuscript and contributed to the final version.

*Competing interests.* The authors declare that they have no conflict of interest

*Acknowledgements.* This work was supported by Climate-KIC through project "SAFERPLACES – Improved assessment of pluvial, fluvial and coastal flood hazards and risks in European cities as a mean to build safer and resilient communities", Task ID TC2018_B4.7.3-

SAFERPL_P430-1A KAVA2 4.7.3. Further funding was received funding from the European Union's Horizon 2020 research and innovation programme under grant agreement no. 730381. The authors would like to thank Dennis Wagenaar (Deltares) for kindly sharing the data from the 1993 Meuse flood, the office of the Polish surveyor-general for providing topographical data from the national cartographic repository, and colleagues at GFZ German Research Centre for Geosciences for their help with extracting the flood damage data contained in the HOWAS21 database (http://howas21.gfz-potsdam.de/howas21/). We also thank Danijel Schorlemmer (GFZ German Research Centre for

Geosciences) for technical discussions.



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

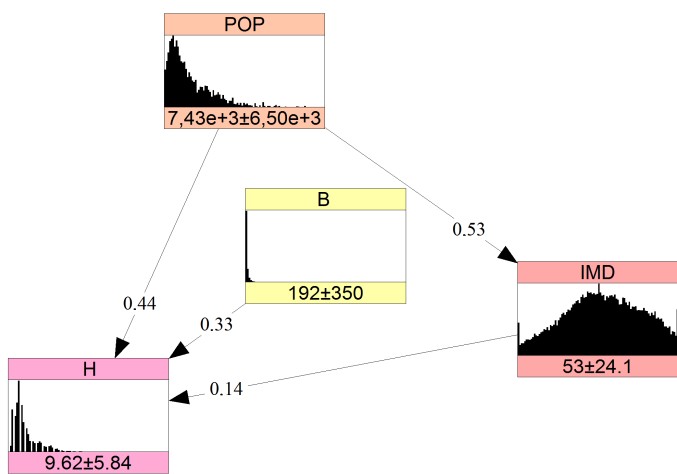

**Figure 1.** A Bayesian Network for predicting residential building height. Values on the arcs represent the (conditional) rank correlation; values under the histograms are the mean and standard deviation of the marginal distributions. H – building height [m], POP – population density, IMD – soil sealing [%], B – building footprint area [m2]. Graph generated using Uninet software (Hanea et al., 2015).

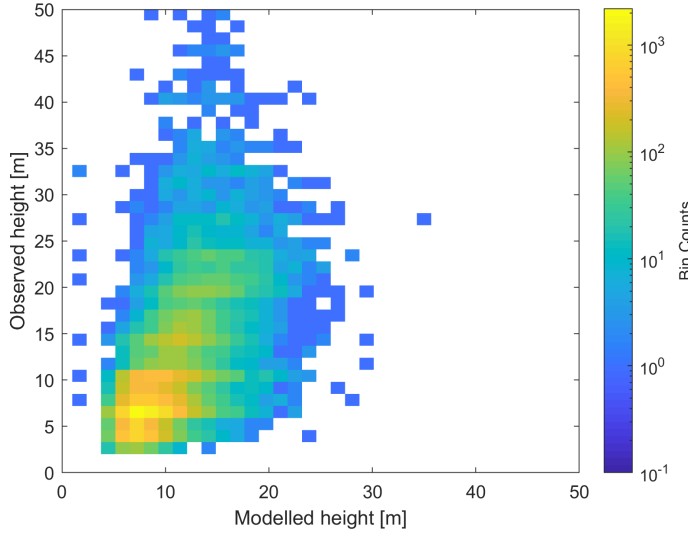

**Figure 2.** Binned scatter plot for observed and modelled heights of residential buildings for 30 European capitals, out-of-sample validation.





**Figure 3.** Value of (a) residential buildings and (b) household contents per m$^2$ of floor space, per dwelling/household and per person, ranked by values per m$^2$ of floor space, as of 2017.
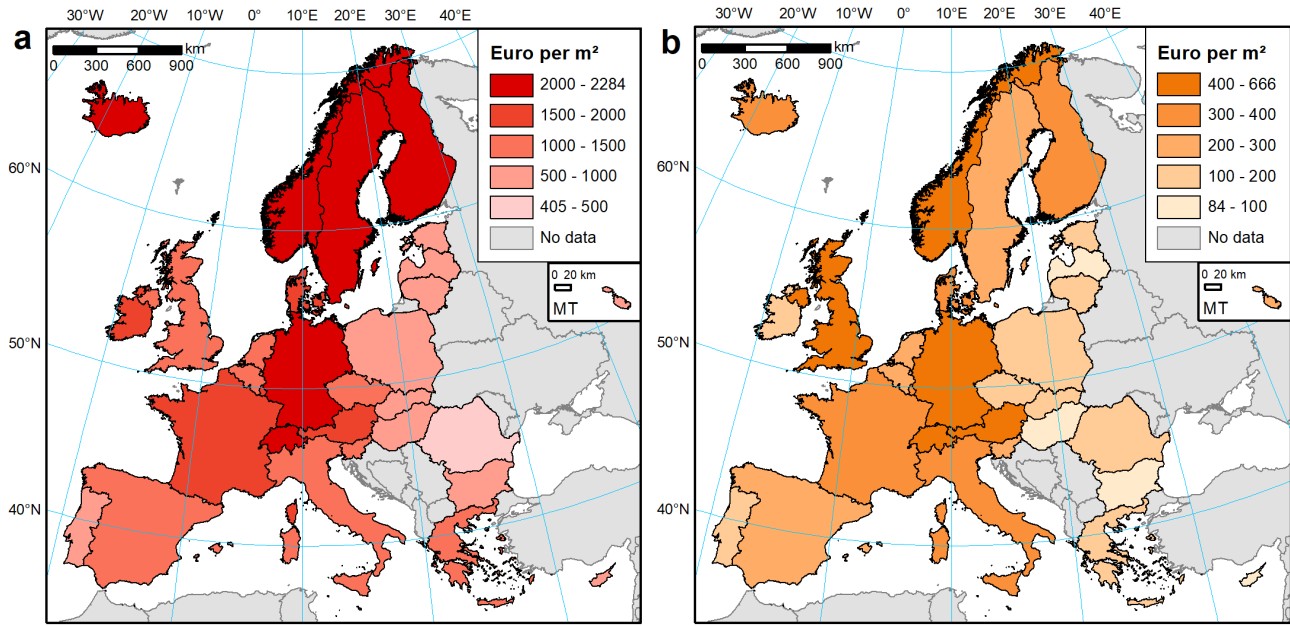

**Figure 4.** Value of (a) residential buildings and (b) household contents per m$^2$ of floor space as of 2017. Country boundaries from EuroGeographics (Eurostat, 2019b).

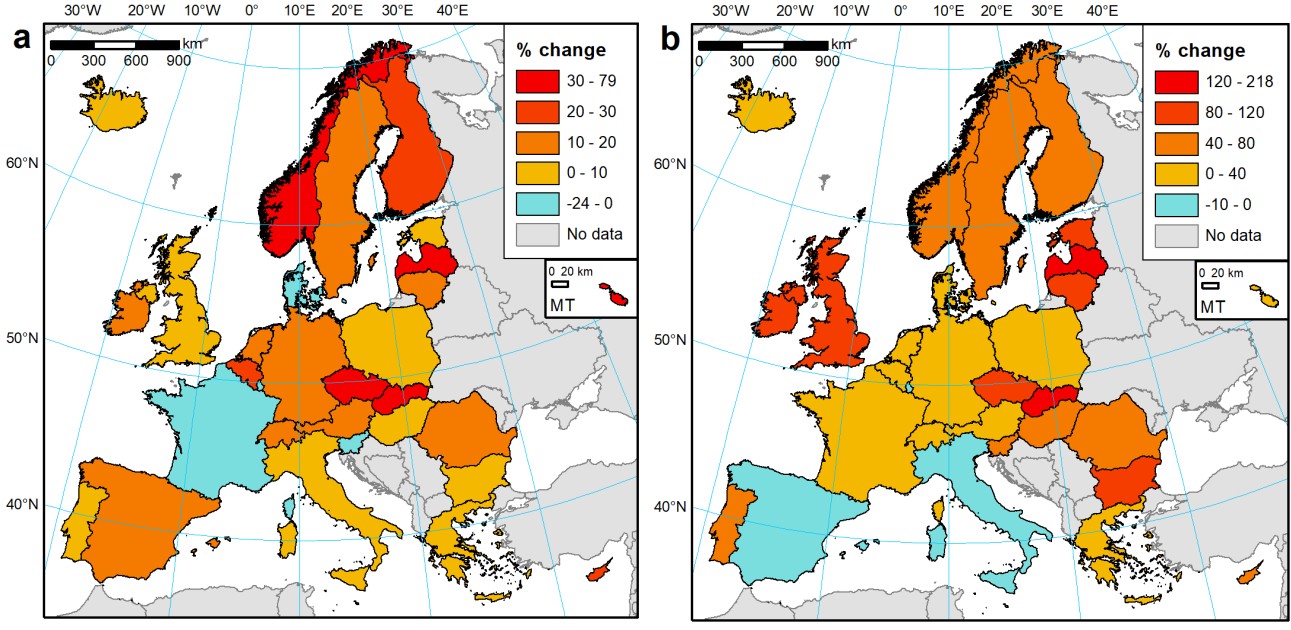

**Figure 5.** Change in the value of (a) residential buildings and (b) household contents per m$^2$ of floor space, 2000–2017, in constant prices. Country boundaries from EuroGeographics (Eurostat, 2019b).


**Figure 6.** Estimated residential asset values in a low-lying part of the city of Szczecin, Poland. Flood hazard zone from Paprotny and Terefenko (2017). Building geometry from ©OpenStreetMap contributors 2019. Distributed under a Creative Commons BY-SA License.



**Figure 7.** Comparison of residential building values per m² of floor space estimated in this study with (a) two estimates by the Joint Research Centre (Huizinga et al., 2017) for year 2010 and (b) estimates from NERA project (Ozcebe et al., 2014) for year 2011.




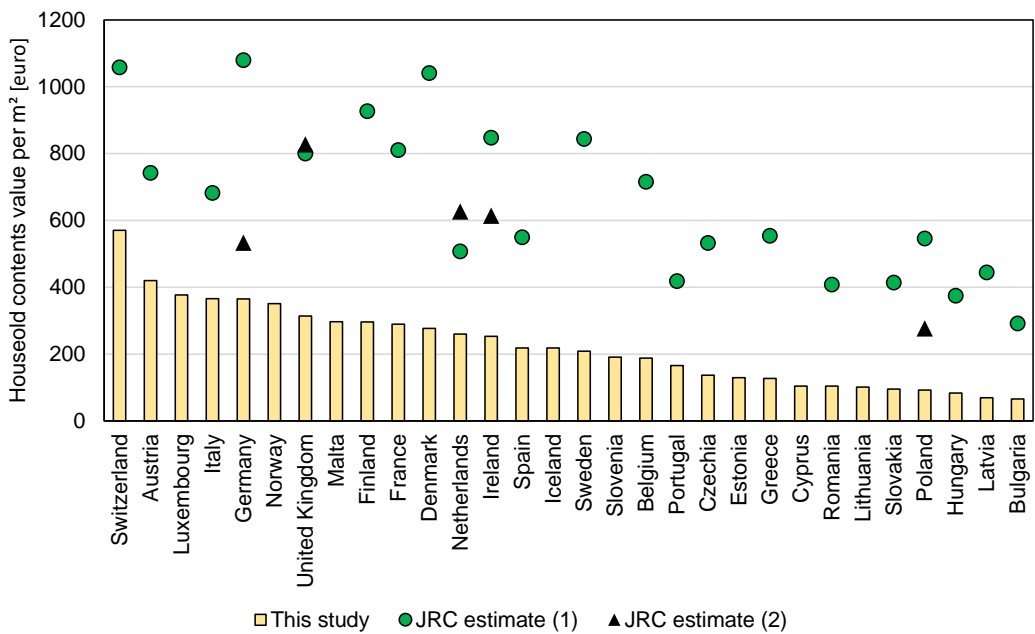

**Figure 8.** Comparison of household contents values per m² of floor space estimated for the year 2010 in this study with two estimates by the Joint Research Centre (Huizinga et al., 2017).

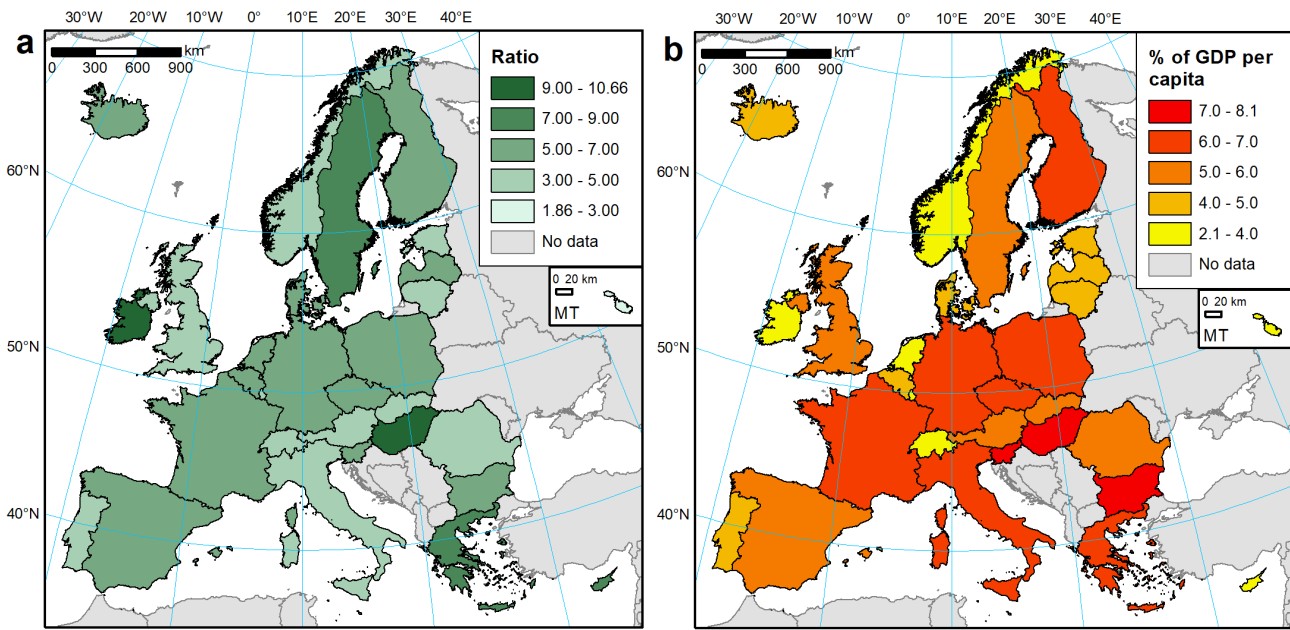

**Figure 9.** (a) Ratio between average building structure and household contents value per m², 2017; (b) Total residential assets (building and contents) as % of GDP per capita, 2017. Country boundaries from EuroGeographics (Eurostat, 2019b).



**Table 1.** Variables considered for the building height prediction model. Abbreviations are shown for variables included in the final model (Fig. 1).

| Variable | Dataset | Source |
|---|---|---|
| Building height (H) [m] | Building Height 2012 | Copernicus Land Monitoring Service (2019) |
| Population per 1 km grid cell (2011 census) (POP) | GEOSTAT 2011 | Eurostat (2019b) |
| Population per 100 m grid cell (2011 census) | HANZE database | Paprotny et al. (2018a) |
| Population in an urban cluster (2011 census) | Urban Clusters 2011 | Eurostat (2019b) |
| Distance from centre of an urban cluster [km] | Urban Clusters 2011 | Eurostat (2019b) |
| Soil sealing per 100 m grid cell (IMD) [%] | Imperviousness 2012 | Copernicus Land Monitoring Service (2019) |
| Build-up surfaces per 100 m grid cell [%] | European Settlement Map 2012 | Copernicus Land Monitoring Service (2019) |
| Building footprint area (B) [$m^2$] | OpenStreetMap | OpenStreetMap (2019) |

**Table 2.** Validation statistics for the building height prediction model (mean value of the uncertainty distribution) for various sets of residential buildings.

| Dataset | N | $R^2$ | MAE | MBE | SMAPE | Obs. mean |
|---|---|---|---|---|---|---|
| Residential building heights, 30 European capitals | 24,212 | 0.36 | 3.24 m | 0.08 m | 0.17 | 9.57 m |
| Number of floors in residential buildings, Polish coastal zone | 62,580 | 0.33 | 0.64 | -0.07 | 0.16 | 2.01 |
| *of which*: houses with 1 flat | 54,410 | 0.13 | 0.57 | -0.10 | 0.16 | 1.80 |
| houses with 2 flats | 1145 | 0.05 | 0.62 | -0.05 | 0.16 | 2.02 |
| houses with 3 or more flats | 7025 | 0.17 | 1.19 | 0.12 | 0.16 | 3.66 |
| Floor space area, detached houses, Meuse flood 1993 | 3043 | 0.42 | 53.8 $m^2$ | -17.3 $m^2$ | 0.18 | 159 $m^2$ |
| Floor space area, all houses, German floods 2002–2014 | 2330 | 0.34 | 122 $m^2$ | 32.3 $m^2$ | 0.26 | 219 $m^2$ |
| *of which*: detached houses | 1351 | 0.15 | 92.1 $m^2$ | 31.1 $m^2$ | 0.33 | 167 $m^2$ |
| semi-detached houses | 443 | 0.20 | 112 $m^2$ | 50.0 $m^2$ | 0.26 | 193 $m^2$ |
| multi-family houses | 536 | 0.31 | 207 $m^2$ | 20.5 $m^2$ | 0.26 | 382 $m^2$ |

**Table 3.** Hit rate of predictions of the number of floors for Polish residential buildings at risk of sea level rise and coastal floods.

| % of correctly predicted floors | | Predicted number of floors | | | | | | | | N |
|---|---|---|---|---|---|---|---|---|---|---|
| | | 1 | 2 | 3 | 4 | 5 | 6 | 7+ | Total | |
| Observed number of floors | 1 | **69.5** | 25.3 | 3.9 | 1.0 | 0.2 | 0.0 | 0.0 | 100.0 | 18178 |
| | 2 | 38.7 | **42.9** | 14.5 | 3.2 | 0.6 | 0.1 | 0.0 | 100.0 | 32325 |
| | 3 | 10.1 | 48.8 | **25.1** | 9.6 | 5.1 | 1.2 | 0.2 | 100.0 | 8227 |
| | 4 | 1.9 | 18.2 | 25.0 | **24.8** | 18.1 | 8.1 | 3.9 | 100.0 | 2161 |
| | 5 | 0.2 | 5.5 | 14.7 | 29.5 | **26.3** | 14.9 | 8.8 | 100.0 | 1337 |
| | 6 | 0.9 | 5.4 | 8.0 | 18.8 | 30.4 | **24.1** | 12.5 | 100.0 | 112 |
| | 7+ | 0.0 | 0.4 | 11.3 | 22.5 | 31.3 | 20.4 | **14.2** | 100.0 | 240 |