# Peer review of "Estimating exposure of residential assets to natural hazards in Europe using open data"

_Natural Hazards and Earth System Sciences, 2019_

## Referee Comment (RC1) · Anonymous Referee #1 · 14 Oct 2019

This paper presents an effort to estimate exposure of residential assets using publicly available or open source data. The rationale behind the choice of data sets is sound and the effort in preparing these different data sets is appreciated. Overall, the undertaking is commendable, since this approach (and resulting estimates) might be of importance for risk and vulnerability analysis in a broad context.

While I think that the approach itself as well as the data sets used are very interesting and promising, I have some concerns and suggestions with respect to the methodology. In general, the methodology section should be reworked to contain more precise, in-depth information on how the authors solved the given task from a methodological point of view. While section 2 is quite long, the methodology is sometimes not very clear, and some parts are quite verbose. Also, I have the impression that the full potential of the data set is not exploited. Results are not ultimately convincing given the mediocre model quality as shown in the validation.

Please find detailed comments on the respective sections and subsections below.

**2.1 Identification of residential buildings**

I agree that *'the problem of accurately identifying buildings and occupancy, especially with open data, is outside the scope of this paper'*. However, it remains unclear how residential buildings were eventually defined in this study. This needs to be clearly stated for the sake of reproducibility. Apparently, two OSM layers (buildings and land use) were downloaded (On a sidenote: a date indicating the day of the download would be nice to reference the status/version of the data set used). Was information obtained from the buildings layer enhanced or modified based on the land use? If so, how?

**2.2 Building size estimation**

- Seven potentially important variables were initially defined. Three of these variables were included in the final model. Even though it can be guessed how these variables were selected (p.4, l.113ff), the variable selection process is not clearly described.

- I think that the use of a 2% sample is somewhat critical, since a lot of information is dropped. Why were so many instances dropped, how was this number (2%) chosen, and how can the authors guarantee that this is a representative sample? The full data set should include roughly 2,373,300 records (2% correspond to 47,466 records). A data frame with 2 million rows and maybe 10 columns is definitely still manageable on local machines.

[Figure]

- In addition, the 2% sample was only used once. Results were then tested once on a 1% sample. This approach is not very robust. Proper $k$-fold cross-validation using the full data set would be desirable.

- What was the reason to use a BN for predicting exposure? Was the BN the only model that was tested, or was it contrasted to other approaches? Once the full data set is created, model comparison is comparatively less time-consuming than data preparation. Since Bayesian approaches are often computationally demanding, a classical regression approach or simple machine learning model (e.g. random forest) might be worth trying. This would also allow to investigate more complex interactions between variables as well as non-linear effects.

- The authors assume that there are no country-specific differences in $H$, apart from those that are implicitly modelled by including *POP*, *IMD* and *B*. The authors claim that they provide a 'universal method for estimating exposure of residential assets' (p.1, l.3f) across whole Europe. Since the method was only validated with data from Poland, Germany and the Netherlands, I am not sure if this statement is fully justified. Since the characteristics might be different in different countries, using a variable specifying geographical location (e.g. country or even broader geographical region) might be helpful to tackle unobserved heterogeneity.

- I found the explanation for the empirical relationship given in Eq. (1) a little bit difficult to understand, since the numbers are scattered throughout the paragraph below the formula. I suggest to streamline this explanation.

- Also, I realized that within Eq. (1), $B$ is used (1.) to derive $H$, and (2.) to compute $F$, which is based on $H$. I don't think that this is a problem, but I noticed that this puts quite a lot of weight on $B$.

**2.3 Valuation of buildings and household contents**

- I suggest to include a supplementary table to show which formula for deriving $S_t$ was used for each country.

**2.4 Validation**

- Generally speaking, the coefficient of determination denotes the share of explained variance in the dependent variable that is predictable using independent variable. Note that $R^2 == r^2$ holds only in special cases such as simple linear regression if an intercept is included. While this is the case in the assessment of predicted vs observed values presented in the paper, where the coefficient of determination equals the square of the correlation coefficient, the authors may want to clarify this.

- Being a very common error metric, root mean squared error could be included as well, since it provides more information content with respect to outliers.

- The first two sentences of Section 2.4.2 are unclear to me. The collective out-of-sample validation was done using an unseen 1% sample across all cities. How was the individual validation performed? By using stratified 1% samples of each city? The second sentence starting with 'Then' suggests that the procedure is different and that the samples are not the same. If the same stratified sample is used, validation results can be assessed both city-specific and at an aggregated European level.

**3 Results**

- An overall $R^2$ of 0.36 is moderate, indeed. This means that only a third of the observed variance in building height can be explained using modelled building height (given that observed vs. predicted regression was used). The confusion matrix (Table 3) showing around 25% (and an increasingly lower amount as the number of floors increases) correctly classified outcomes for buildings with more than 2 floors is also slightly puzzling. Again, this might be a hint to try (1.) using more data and (2.) comparing different modelling approaches. Good results for *average* height are of rather limited explanatory power in terms of model quality assessment, since I would naturally assume that the differences in means are not too large when using any reasonable model. The problem of low variance might also be tackled by (1.) and (2.) mentioned in the previous sentence. That the model does not perform satisfactory at all for cities like Nicosia and Reykjavik might indicate that there are country-specific differences. All cities that exhibit good performance are located in Central Europe (Vienna, Berlin, Amsterdam, Luxembourg, Warsaw, Zagreb).

- In the abstract, a validation with (1) buildings in Poland and (2) a sample of Dutch and German houses is mentioned. In the paper, (1) can be found in section 3.3, and (2) is described in the last paragraph of 3.1. I think the title of subsection 3.3 should be reworked, as 'Example application' is rather generic. Maybe a dedicated validation subsection for these new data sources could be helpful?

- In fact, there does seem to be a slight systematic bias in the results. Figure 2 shows overestimation for low building heights and underestimation of high building heights, with accurate results around 12 m. The regression line likely has a negative intercept and a slope larger than 1.

**4 Discussion**

- The structure of the discussion is generally well thought through. However, the authors again solely focus on the BN model. Maybe the use of other models might lead to better results on the same data set? Limitations of the BN model itself and implications of using a comparatively small sample size (given available data) are not discussed.

**Figures**

- Figure 1: The histogram plots do not have any axis labels and units, which is a major limitation (in terms of information content) of this figure, since the histograms are essentially incomplete. Also - for the sake of consistency: the unit for population density is missing in the caption.

- Figure 2: Please use the same spacing for axis ticks (either steps of 5 or 10).

- Figure 3: I suggest to use points instead of bars. The information that needs to be transported is the value at the end of the bar, not the area of bar itself. Therefore, information density is higher when using points. Also, the two colors of the bars are different (orange indicating building value in a and yellowish indicating household contents value in b), but the legend matches only the color in b.

- Figures 4 & 5: I think it should be mentioned in the caption that values for each country are based on the respective capitals, since this is important when interpreting the results.

- Figure 7: Legend for a is missing, only legend for b is provided. Again, I suggest to consider using points instead of bars. If points overlap, you may slightly jitter them along the x axis or use some transparency.

- Figure 8: It seems that this figure is not referenced in the text. If this is the case (I might have overlooked it), please add a reference in the text. Also, it reveals a substantial differences when compared to the JRC values, this could be explored/discussed further. Again, I suggest to consider using points instead of bars.

- Figure 9: It seems that this figure is not referenced in the text. If this is the case (I might have overlooked it), please add a reference in the text.

**Tables**

- Table 1: I am wondering why two different sources for 'Population per area' were used. If both are based on the 2011 census, why not using the one with higher resolution if the model is fitted at a building level?

- Table 3: '% of correctly predicted floors' is confusing. Only the diagonal values indicates the percentage of *correctly* predicted floors, all other number are simply the percentage of predicted floors?

**Formal aspects**

- p.7: $L_m ean$ should read $L_{mean}$, or simply $\bar{L}$.

- Please check consistency regarding capitalization (e.g.: 'Eq.' vs 'eq.'). NHESS manuscript preparation instructions suggest 'Eq.'.

- Please format the supplement according to the journal's standards.

**Final remark**

To conclude: Given the mediocre performance, I suggest (1) trying to use the full data set instead of the 2% sample only, and (2) taking a look at alternative modelling approaches for comparative purposes. If this does not improve the results, the input data might indeed be of limited use for predicting building height, but it does not leave the reader wondering the potential of the full data set has not been explored. Since the effort of gathering and preparing this interesting data set has already been undertaken, I suggest to have another look on the readily compiled data set. I am in full support of publishing these findings, but I propose reconsideration after major revision.

---

## Referee Comment (RC2) · Anonymous Referee #2 · 18 Oct 2019

The manuscript describes a compilation of available datasets useable for deriving principal data for exposure analyses. Tthe authors show exemplarily the development of a Bayesian Network-based model for predicting building heights. The manuscript provides a very usefull overview of prospective dataset sand methods for assessing the values at risk in exposure and risk analyses. Thus, I recommend the publication of the manuscript. However, the manuscript has to be improved first in terms of readability.

One of my main concerns is the rather unorganized structure of the manuscript. It deals with different scales, e.g. nationally aggregated data, data from 30 major cities, different validation samples, and a local case study. Moreover, the manuscript has also different time scales. It lists different states of the datasets and includes timeseries of the temporal development of the economic values. This is on the one side a benefit in

terms of the broad scope but hampers the readability for the reader on the other side. I urge the authors to elaborate a more thorough structure of the manuscript to help reader's orientation.

Another concern is the transferability of the model from an urban context to a rural context. This needs to be validated. For instance, Roethlisberger et al (2018) in the same journal state that a difference in the values per square meter between Centre areas (urban areas) and Residential areas of 60%. An alternative is to restrict the title of the manuscript towards urban context of Europe. The conclusion in the abstract "The study shows that the resulting standardized residential exposure values provide much better coverage and consistency compared to previous studies" is not supported by the results.

In the Introduction section, the authors state that the developed procedure is "applicable in any location". This proof is not provided (discussion about urban-rural context). Another main criticism is that the identification of residential buildings is not described. There is an explicit subsection on this topic (section "2.1 Identification of residential buildings"). However, how this identfication has been done is described within brackets in section "2.2 Building size estimation" in line 109 ("(identified either through the buildings or the land use layers of OSM)") while in section 2.1 is stated that the identification of buildings and their occupancy (i.e., residential use?) is outside the scope of this paper.

However, I am looking forward to the publication of an improved version of this manuscript.

---

## Author Comment (AC1) · 11 Nov 2019

We thank the reviewer for taking the time to analyse our manuscript. Below we list the comments (C) and our responses (R).

C: I agree that 'the problem of accurately identifying buildings and occupancy, especially with open data, is outside the scope of this paper'. However, it remains unclear how residential buildings were eventually defined in this study. This needs to be clearly stated for the sake of reproducibility. Apparently, two OSM layers (buildings and land use) were downloaded (On a sidenote: a date indicating the day of the download would be nice to reference the status/version of the data set used). Was information obtained from the buildings layer enhanced or modified based on the land use? If so, how?

[Figure]

R: Firstly, we downloaded two Map Features ("Buildings" and "Landuse") for the 30 study areas and the example application study from section 3.3. For the analysis, residential buildings were objects from "Buildings" layer which (1) had tags "residential", "apartments", "house", "detached" or "terrace", and (2) had tags "yes" (indicating that a building exists, but the function is not defined) and were located within an object from "Landuse" layer which had a tag "residential". OSM is updated continuously, and the data used here were downloaded between 22 and 25 January 2019. Data for the example application was downloaded from OSM on 18 July 2019. We will add this information to the text.

C: Seven potentially important variables were initially defined. Three of these variables were included in the final model. Even though it can be guessed how these variables were selected (p.4, l.113), the variable selection process is not clearly described.

R: The variables first variable (POP) was chosen as it had the highest unconditional rank correlation with H (Table S1). The second variable (B) had the highest conditional rank correlation with H among the remaining six variables, and then IMD had the highest conditional rank correlation with H among the remaining five variables. Further variables had very low (<0.1) correlation with H only, therefore only three variables were used to explain H. Remaining arc between POP and IMD was added due to high correlation between the two. All arcs further have a theoretical explanation, as described in lines 117-122. We will clarify the explanation in the text.

C: I think that the use of a 2% sample is somewhat critical, since a lot of information is dropped. Why were so many instances dropped, how was this number (2%) chosen, and how can the authors guarantee that this is a representative sample? The full data set should include roughly 2,373,300 records (2% correspond to 47,466 records). A data frame with 2 million rows and maybe 10 columns is definitely still manageable on local machines.

R: The principle reason for using a sample of the OSM buildings for all cities was reducing the time of data processing with GIS to arrive with table of data. Then, the efficiency of a non-parametric Bayesian Network calculation (or a Random Forest computation) drops significantly with large numbers of data points, which would make it less suitable for a possible operational pan-European application. Those aspects notwithstanding, the buildings of 30 capital regions are not the complete population of the European residential buildings, but a small fraction of them, and using all data would still mean using a sample of European buildings. We went back to the full dataset and extracted a larger share of the data, so that the influence of using different dataset size can be presented. The whole population of usable data (i.e. no data missing for any variable at a given location) results in a correlation matrix almost identical correlation matrix, differing by a rank correlation of 0.007 at most. This matrix (N = 2,375,058) will be used to revise Table S1.

C: In addition, the 2% sample was only used once. Results were then tested once on a 1% sample. This approach is not very robust. Proper k-fold cross-validation using the full data set would be desirable.

R: We made a new extraction of the data from the full dataset, containing 10% of the data. This sample was used for a 10-fold cross-validation, and the revision of other results. We have revised Tables 2 and 3 (see supplement to this response), and we will revise related text accordingly.

C: What was the reason to use a BN for predicting exposure? Was the BN the only model that was tested, or was it contrasted to other approaches? Once the full data set is created, model comparison is comparatively less time-consuming than data preparation. Since Bayesian approaches are often computationally demanding, a classical regression approach or simple machine learning model (e.g. random forest) might be worth trying. This would also allow to investigate more complex interactions between variables as well as non-linear effects.

R: There are several advantages of the class of BNs used here, compared with other

approaches: (1) they are probabilistic, providing uncertainty bounds, (2) they can be applied when some of the input data is missing (which Random Forests can't), as e.g. the imperviousness dataset is not gap-free for Europe, which makes it more useful for applications, (3) the whole model can be presented graphically (in Random Forests, only singular trees out of the forest could be presented), (4) accuracy depends only on the configuration of nodes and arcs, and not on model parameters such as number of splits, leaves, trees etc. We applied Random Forests to our data (using Matlab functions 'TreeBagger' and 'predict'), with a 10-fold cross-validation on the basis of the 10% of the data – the same set-up as for the BN. The number of trees was set to 100, maximum number of leaves to 50, in-bag fraction to 1/3, and splits to two. The resulting R2 is slightly lower, RMSE higher, MBE strongly negative and only MAE is slightly lower than when using the BN. Overall, the performance is quite similar, though still worse. The comparative results were added to the supplement of this response.

C: The authors assume that there are no country-specific differences in H, apart from those that are implicitly modelled by including POP, IMD and B. The authors claim that they provide a 'universal method for estimating exposure of residential assets' (p.1, l.3f) across whole Europe. Since the method was only validated with data from Poland, Germany and the Netherlands, I am not sure if this statement is fully justified. Since the characteristics might be different in different countries, using a variable specifying geographical location (e.g. country or even broader geographical region) might be helpful to tackle unobserved heterogeneity.

R: The datasets for Poland, Germany and the Netherlands were the only independent datasets with information on floor space or number of floors for individual buildings that we were able to collect. If the reviewer knows other datasets like those for other countries, we would add them to the analysis. Regarding the heterogeneity, individual models for each country could be made, and they would improve performance at each, but this would still give only a good prediction for the capital city region of a given country, leaving uncertainty how well such a model would perform in other parts of the

country. We will move results for individual cities from the Supplement to the main text to make the variation in the model's quality more visible to the reader (see supplement to this response). We also note that we discovered and corrected an error related to the dataset German flood-affected households. In the submission, only those households affected by fluvial floods were included, and not those affected by pluvial floods as suggested by the years provided in Table 2. We have added those missing households, increasing the number of records from 2330 to 2868, though the validation results didn't change much.

C: I found the explanation for the empirical relationship given in Eq. (1) a little bit difficult to understand, since the numbers are scattered throughout the paragraph below the formula. I suggest to streamline this explanation.

R: We will clarify the explanation in the paragraph.

C: Also, I realized that within Eq. (1), B is used (1.) to derive H, and (2.) to compute F, which is based on H. I don't think that this is a problem, but I noticed that this puts quite a lot of weight on B.

R: The conditional correlation between B and H was considerable, therefore it had to be included in predicting H, and it is indispensable in calculating F, which strongly depends on both B and H.

C: I suggest to include a supplementary table to show which formula for deriving St was used for each country.

R: A supplementary table will be added (see table X in the supplement to this response).

C: Generally speaking, the coefficient of determination denotes the share of explained variance in the dependent variable that is predictable using independent variable. Note that $R2 == r2$ holds only in special cases such as simple linear regression if an intercept is included. While this is the case in the assessment of predicted vs observed

values presented in the paper, where the coefficient of determination equals the square of the correlation coefficient, the authors may want to clarify this.

R: The linear regression of revised Fig. 1 is approximately 1.02x − 0.3 (varies slightly within the 10-fold cross-validation).

C: Being a very common error metric, root mean squared error could be included as well, since it provides more information content with respect to outliers.

R: RMSE will be added to the results.

C: The first two sentences of Section 2.4.2 are unclear to me. The collective out-of-sample validation was done using an unseen 1% sample across all cities. How was the individual validation performed? By using stratified 1% samples of each city? The second sentence starting with 'Then' suggests that the procedure is different and that the samples are not the same. If the same stratified sample is used, validation results can be assessed both city-specific and at an aggregated European level.

R: We replaced the existing results with a 10-fold cross-validation (see attached revised tables).

C: An overall R2 of 0.36 is moderate, indeed. This means that only a third of the observed variance in building height can be explained using modelled building height (given that observed vs. predicted regression was used). The confusion matrix (Table 3) showing around 25% (and an increasingly lower amount as the number of floors increases) correctly classified outcomes for buildings with more than 2 floors is also slightly puzzling. Again, this might be a hint to try (1.) using more data and (2.) comparing different modelling approaches. Good results for average height are of rather limited explanatory power in terms of model quality assessment, since I would naturally assume that the differences in means are not too large when using any reasonable model. The problem of low variance might also be tackled by (1.) and (2.) mentioned in the previous sentence. That the model does not perform satisfactory at all for cities

like Nicosia and Reykjavik might indicate that there are country-specific differences. All cities that exhibit good performance are located in Central Europe (Vienna, Berlin, Amsterdam, Luxembourg, Warsaw, Zagreb).

R: We added more data (10% instead 2%), but this had only marginal effect on the data, as one might expect from a randomized sample of a very large dataset. Several cities have rather poor performance, but it is also partly due to variation in the quality of height data (which routinely shows errors of 1–3 m, according to the validation information contained in the dataset) and OSM buildings (which is particularly poor for Nicosia, for instance, using visual inspection). Some of the cities with good results are not located in Central, but Western and Northern Europe (Amsterdam, Berlin, Luxembourg, Stockholm, Vienna). We revisited the dataset using Random Forests, as described in a previous comment, which didn't lead to improvement. However, we tested the model for different urban-rural typologies using "Degrees of Urbanisation 2014" dataset from Eurostat. This allows classifying our data points according to whether they were located in "cities", "towns and suburbs" or "rural areas" (defined at the level of local administrative units - LAUs). The results presented in a table in the supplement to this response show that the R2 is lower for towns and rural areas than for cities, though this stems from the much lower variation in building height; MAE is lower in those cases, and in all three types of LAUs MAE has almost the exact same proportion to average height.

C: In the abstract, a validation with (1) buildings in Poland and (2) a sample of Dutch and German houses is mentioned. In the paper, (1) can be found in section 3.3, and (2) is described in the last paragraph of 3.1. I think the title of subsection 3.3 should be reworked, as 'Example application' is rather generic. Maybe a dedicated validation subsection for these new data sources could be helpful?

R: The example in section 3.3 is not based on the validation dataset for Poland. The Polish dataset is from a government-run national database (BDOT) and is used for validation in 3.1 as shown in Tables 2 and 3. The example in 3.3 uses an extraction of OSM data, as the BDOT dataset is not openly available in contrast to OSM. The

example is meant to present how the methods from the paper can be used in practice. We will highlight this better in the text.

C: In fact, there does seem to be a slight systematic bias in the results. Figure 2 shows overestimation for low building heights and underestimation of high building heights, with accurate results around 12 m. The regression line likely has a negative intercept and a slope larger than 1.

R: The linear regression of revised Fig. 1 is approximately 1.02x – 0.3 (varies slightly within the 10-fold cross-validation). We note that the model underestimates the height of tall buildings, but this is partly because few buildings are very tall.

C: The structure of the discussion is generally well thought through. However, the authors again solely focus on the BN model. Maybe the use of other models might lead to better results on the same data set? Limitations of the BN model itself and implications of using a comparatively small sample size (given available data) are not discussed.

R: We will add more information about the uncertainty related to the data analysis, and add results generated with Random Forests.

C: Figure 1: The histogram plots do not have any axis labels and units, which is a major limitation (in terms of information content) of this figure, since the histograms are essentially incomplete. Also - for the sake of consistency: the unit for population density is missing in the caption.

R: The graph and caption will be corrected.

C: Figure 2: Please use the same spacing for axis ticks (either steps of 5 or 10).

R: The ticks will be corrected.

C: Figure 3: I suggest to use points instead of bars. The information that needs to be transported is the value at the end of the bar, not the area of bar itself. There-

fore, information density is higher when using points. Also, the two colors of the bars are different (orange indicating building value in a and yellowish indicating household contents value in b), but the legend matches only the color in b.

R: The legend will be corrected and the graph will be reworked, so that points are used instead of bars.

C: Figures 4 & 5: I think it should be mentioned in the caption that values for each country are based on the respective capitals, since this is important when interpreting the results.

R: The values for each are not based on capitals, but on the economic statistics at national level (section 2.3).

C: Figure 7: Legend for a is missing, only legend for b is provided. Again, I suggest to consider using points instead of bars. If points overlap, you may slightly jitter them along the x axis or use some transparency.

R: The legend will be corrected and the graph will be reworked, so that points are used instead of bars.

C: Figure 8: It seems that this figure is not referenced in the text. If this is the case (I might have overlooked it), please add a reference in the text. Also, it reveals a substantial difference when compared to the JRC values, this could be explored/discussed further. Again, I suggest to consider using points instead of bars.

R: We incorrectly refer to Fig. 7 instead of Fig. 8 in line 410. The graph will be reworked, so that points are used instead of bars. The large difference between JRC and our estimates could be caused by the assumption of a single ratio between building and contents loss (which we show is far from uniform), transposition of this value from an American flood damage model, and possible differences in definition (JRC estimate including more items). We will describe this aspect better in the revision.

C: Figure 9: It seems that this figure is not referenced in the text. If this is the case (I

might have overlooked it), please add a reference in the text.

R: Fig. 9 should have been mentioned in lines 413 and 532.

C: Table 1: I am wondering why two different sources for 'Population per area' were used. If both are based on the 2011 census, why not using the one with higher resolution if the model is fitted at a building level?

R: the 1 km data are, in most cases, an aggregation of georeferenced records of all enumerated population during the 2011 censuses, therefore represent very accurately the spatial distribution of population. The 100 m dataset is a disaggregated version constructed in the cited study using land cover/use and soil sealing data, therefore introducing modelling error. The 1 km dataset also represents neighbourhood/urban district population, and the 100 m the population of a group of buildings, and the first proved more relevant to represent the dominant type of buildings in an area.

C: Table 3: '% of correctly predicted floors' is confusing. Only the diagonal values indicates the percentage of correctly predicted floors, all other number are simply the percentage of predicted floors?

R: Yes, the text in the table should be different, we will change it to "% of predicted floors within observed floor class"

C: p.7: Lmean should read Lmean, or simply LÂŕ.

R: This will be corrected to L_{mean}.

C: Please check consistency regarding capitalization (e.g.: 'Eq.' vs 'eq.'). NHESS manuscript preparation instructions suggest 'Eq.'.

R: We will correct this in the manuscript.

C: Please format the supplement according to the journal's standards.

R: The supplement will be reformatted according to the NHESS Word template.

Please also note the supplement to this comment:
https://www.nat-hazards-earth-syst-sci-discuss.net/nhess-2019-313/nhess-2019-313-AC1-supplement.pdf

[Figure]

**Supplement:**

*Table 1. Comparison of average results from a 10-fold cross-validation.*

| Method | N | $R^2$ | MAE (m) | MBE (m) | SMAPE | RMSE (m) | Observed mean (m) |
|---|---|---|---|---|---|---|---|
| NPBN | 23,736 | 0.35 | 3.25 | 0.09 | 0.17 | 4.72 | 9.60 |
| Random Forest | 23,736 | 0.30 | 3.11 | -1.40 | 0.18 | 5.26 | 9.60 |

*Table 2. Validation statistics for the building height prediction model (mean value of the uncertainty distribution), depending on the degree of urbanization at municipal level. Validated for 2% of data points in cities and towns/suburbs, and for all data points in rural areas.*

| Degree of urbanisation | N | $R^2$ | MAE (m) | MBE (m) | SMAPE | RMSE (m) | Observed mean (m) |
|---|---|---|---|---|---|---|---|
| Cities | 44,949 | 0.35 | 3.32 | 0.06 | 0.17 | 4.79 | 9.77 |
| Towns and suburbs | 2685 | 0.15 | 2.15 | 0.63 | 0.14 | 2.85 | 7.13 |
| Rural areas | 3191 | 0.24 | 2.09 | 0.34 | 0.16 | 3.44 | 6.44 |

*Table 3. Validation statistics for the building height prediction model (mean value of the uncertainty distribution) for different cities. For all cities, the results are an average of results for a 10-fold cross-validation. For individual cities, the results are an out-of-sample validation (i.e. the model's sample excluded the city that was validated).*

| Area | N | R^2 | MAE [m] | MBE [m] | SMAPE | RMSE [m] | Obs. mean [m] |
|---|---|---|---|---|---|---|---|
| All cities (cross-validation) | 23,736 | 0.35 | 3.25 | 0.09 | 0.17 | 4.72 | 9.60 |
| Amsterdam | 24,506 | 0.31 | 2.50 | -0.17 | 0.15 | 3.43 | 8.69 |
| Athens | 18,177 | 0.25 | 4.38 | -1.70 | 0.16 | 5.52 | 14.20 |
| Berlin | 25,526 | 0.49 | 3.65 | -1.29 | 0.18 | 5.10 | 10.51 |
| Bratislava | 926 | 0.42 | 6.81 | -4.61 | 0.30 | 10.44 | 13.57 |
| Brussels | 19,845 | 0.12 | 3.77 | -1.00 | 0.17 | 5.00 | 11.50 |
| Bucharest | 1695 | 0.36 | 6.01 | 1.00 | 0.28 | 7.93 | 10.35 |
| Budapest | 1963 | 0.37 | 4.14 | -1.72 | 0.19 | 6.76 | 11.80 |
| Copenhagen | 10,747 | 0.24 | 2.55 | 2.00 | 0.17 | 3.32 | 6.42 |
| Dublin | 12,648 | 0.09 | 1.69 | 1.13 | 0.12 | 2.21 | 6.57 |
| Helsinki | 8053 | 0.34 | 2.62 | 1.01 | 0.17 | 3.68 | 7.11 |
| Lisbon | 3486 | 0.10 | 5.37 | -0.60 | 0.20 | 7.42 | 13.42 |
| Ljubljana | 1196 | 0.19 | 3.28 | 1.97 | 0.22 | 4.39 | 6.35 |
| London | 22,17 | 0.10 | 3.36 | 2.65 | 0.18 | 4.48 | 7.79 |
| Luxembourg | 582 | 0.19 | 2.26 | -0.11 | 0.12 | 3.18 | 9.60 |
| Madrid | 4909 | 0.13 | 6.19 | -1.43 | 0.20 | 8.72 | 16.22 |
| Nicosia | 283 | 0.05 | 3.23 | -0.70 | 0.18 | 4.60 | 9.23 |
| Oslo | 4750 | 0.45 | 2.76 | 1.68 | 0.18 | 3.52 | 6.79 |
| Paris | 23,441 | 0.23 | 3.03 | 0.99 | 0.16 | 4.60 | 8.98 |
| Prague | 6802 | 0.47 | 3.92 | -1.86 | 0.19 | 6.02 | 11.46 |
| Reykjavik | 2364 | 0.05 | 2.99 | 2.05 | 0.22 | 3.61 | 5.80 |
| Riga | 1423 | 0.29 | 4.31 | -1.57 | 0.21 | 6.53 | 11.10 |
| Rome | 5397 | 0.36 | 3.97 | -1.69 | 0.16 | 5.45 | 13.14 |
| Sofia | 4127 | 0.39 | 4.35 | -0.56 | 0.21 | 6.38 | 10.49 |

| | | | | | | | |
|---|---|---|---|---|---|---|---|
| Stockholm | 8748 | 0.25 | 2.23 | 0.62 | 0.16 | 3.48 | 6.82 |
| Tallinn | 1386 | 0.39 | 3.48 | 0.58 | 0.21 | 5.18 | 8.13 |
| Valletta | 123 | 0.13 | 4.32 | 0.24 | 0.18 | 6.49 | 11.66 |
| Vienna | 8690 | 0.50 | 2.89 | -0.11 | 0.16 | 4.34 | 9.34 |
| Vilnius | 757 | 0.42 | 2.79 | -0.91 | 0.17 | 4.82 | 8.86 |
| Warsaw | 7662 | 0.24 | 3.05 | -0.14 | 0.17 | 5.22 | 9.10 |
| Zagreb | 4979 | 0.17 | 2.58 | 0.31 | 0.16 | 4.07 | 8.05 |

*Table 4. Validation statistics for the building height prediction model (mean value of the uncertainty distribution) for various sets of residential buildings.*

| Dataset | N | R^2 | MAE | MBE | SMAPE | RMSE | Obs. mean |
|---|---|---|---|---|---|---|---|
| Number of floors in residential buildings, Polish coast | 62,58 | 0.33 | 0.65 | -0.06 | 0.16 | 1.02 | 2.01 |
| of which: houses with 1 flat | 54,41 | 0.13 | 0.58 | -0.10 | 0.16 | 0.85 | 1.80 |
| houses with 2 flats | 1145 | 0.04 | 0.64 | -0.04 | 0.16 | 0.95 | 2.02 |
| houses with 3 or more flats | 7025 | 0.16 | 1.24 | 0.18 | 0.16 | 1.86 | 3.66 |
| Floor space area, detached houses, Meuse flood 1993 | 3043 | 0.41 | 54.0 m^2 | -17.3 m^2 | 0.18 | 83.5 m^2 | 160 m^2 |
| Floor space area, all houses, German floods 2002-2014 | 2868 | 0.33 | 119 m^2 | 32.9 m^2 | 0.26 | 206 m^2 | 214 m^2 |
| of which: detached houses | 1556 | 0.15 | 94.5 m^2 | 34.4 m^2 | 0.26 | 138 m^2 | 166 m^2 |
| semi-detached houses | 662 | 0.20 | 100 m^2 | 43.2 m^2 | 0.25 | 147 m^2 | 178 m^2 |
| multi-family houses | 647 | 0.30 | 196 m^2 | 19.3 m^2 | 0.26 | 346 m^2 | 366 m^2 |

[Figure]

*Figure 1. Binned scatter plot for observed and modelled heights of residential buildings for 30 European capitals, out-of-sample validation.*

*Table S3. Reference to methodologies used for obtaining building stock.*

| Method | Countries |
|---|---|
| Building stock taken directly from Eurostat database | Austria, Belgium, Cyprus, Czechia, Denmark, Estonia, Finland, France, Germany, Greece, Hungary, Ireland, Italy, Lithuania, Luxembourg, Netherlands, Portugal, Slovakia, Slovenia, United Kingdom |
| Eq. 3 (PIM) | Iceland, Malta, Norway, Spain, Sweden and Switzerland |
| Eq. 4 (modified PIM) | Bulgaria, Latvia, Poland, Romania |

---

## Author Comment (AC2) · 11 Nov 2019

We thank the referee for his review. Below we list the comments (C) and our responses (R).

C: One of my main concerns is the rather unorganized structure of the manuscript. It deals with different scales, e.g. nationally aggregated data, data from 30 major cities, different validation samples, and a local case study. Moreover, the manuscript has also different time scales. It lists different states of the datasets and includes timeseries of the temporal development of the economic values. This is on the one side a benefit in terms of the broad scope but hampers the readability for the reader on the other side. I urge the authors to elaborate a more thorough structure of the manuscript to help
reader's orientation.

R: We prepared a new graphic with the workflow of the paper, with references to all sections, figures and tables, which will be included at the beginning of the methodology section. We add the figure to the supplement to this response.

C: Another concern is the transferability of the model from an urban context to a rural context. This needs to be validated. For instance, Roethlisberger et al (2018) in the same journal state that a difference in the values per square meter between Centre areas (urban areas) and Residential areas of 60%. An alternative is to restrict the title of the manuscript towards urban context of Europe. The conclusion in the abstract "The study shows that the resulting standardized residential exposure values provide much better coverage and consistency compared to previous studies" is not supported by the results.

R: The referenced paper shows a 60% difference in the insured values per square meter of landuse, not of replacement values per usable floor space as in our paper (Table 2 of Roethlisberger et al. 2018). The other authors' findings are not surprising given the higher density of construction in the city/town centers (to which "Centre" class refers) that in other residential areas. Table 3 of Roethlisberger et al. (2018) actually shows lower average insured values per building volume for "Centre" zones (861 CHF per Mˆ3) than in "Residential" areas (897). However, we agree that there are differences between asset values in urban and rural areas, as exemplified by the cited Portuguese data. We will highlight this aspect better in the discussion, but at the moment the availability of both regional economic data and local validation data is very limited, and therefore we didn't attempt (yet) to calculate sub-national asset values per mˆ2.

C: In the Introduction section, the authors state that the developed procedure is "applicable in any location". This proof is not provided (discussion about urban-rural context). Another main criticism is that the identification of residential buildings is not described.

There is an explicit subsection on this topic (section "2.1 Identification of residential buildings"). However, how this identification has been done is described within brackets in section "2.2 Building size estimation" in line 109 ("(identified either through the buildings or the land use layers of OSM)") while in section 2.1 is stated that the identification of buildings and their occupancy (i.e., residential use?) is outside the scope of this paper.

R: We will add information on the selection of OSM buildings. Firstly, we downloaded two Map Features ("Buildings" and "Landuse") for the 30 study areas and the example application study from section 3.3. For the analysis, residential buildings were objects from "Buildings" layer which (1) had tags "residential", "apartments", "house", "detached" or "terrace", and (2) had tags "yes" (indicating that a building exists, but the function is not defined) and were located within an object from "Landuse" layer which had a tag "residential".

Please also note the supplement to this comment:
https://www.nat-hazards-earth-syst-sci-discuss.net/nhess-2019-313/nhess-2019-313-AC2-supplement.pdf

**Supplement:**

[Figure]

*Figure 1. Workflow of the study. Boxes are coloured according to categories explained in the legend. In the top-left corners of the boxes are references to relevant sections of this paper. In the top-right corners of the boxes are references to figures, tables, supplementary tables (S.Tab.) in Supplementary Information 1, equations and Supplementary Information 2 (S.Inf. 2).*

---

## Author Response (AR1)

We thank the reviewers and editor for taking the time to analyse our manuscript. Below we list the comments (C) and our responses (R). After this response, the marked-up version of the manuscript is included.

**Reviewer 1**

C: I agree that 'the problem of accurately identifying buildings and occupancy, especially with open data, is outside the scope of this paper'. However, it remains unclear how residential buildings were eventually defined in this study. This needs to be clearly stated for the sake of reproducibility. Apparently, two OSM layers (buildings and land use) were downloaded (On a sidenote: a date indicating the day of the download would be nice to reference the status/version of the data set used). Was information obtained from the buildings layer enhanced or modified based on the land use? If so, how?

R: Firstly, we downloaded two Map Features ("Buildings" and "Landuse") for the 30 study areas and the example application study from section 3.3. For the analysis, residential buildings were objects from "Buildings" layer which (1) had tags "residential", "apartments", "house", "detached" or "terrace", and (2) had tags "yes" (indicating that a building exists, but the function is not defined) and were located within an object from "Landuse" layer which had a tag "residential". OSM is updated continuously, and the data used here were downloaded between 22 and 25 January 2019. Data for the example application was downloaded from OSM on 18 July 2019. We added this information to p.4, l.90-96.

C: Seven potentially important variables were initially defined. Three of these variables were included in the final model. Even though it can be guessed how these variables were selected (p.4, l.113), the variable selection process is not clearly described.

R: The variables first variable (POP) was chosen as it had the highest unconditional rank correlation with H (Table S1). The second variable (B) had the highest conditional rank correlation with H among the remaining six variables, and then IMD had the highest conditional rank correlation with H among the remaining five variables. Further variables had very low (<0.1) correlation with H only, therefore only three variables were used to explain H. Remaining arc between POP and IMD was added due to high correlation between the two. All arcs further have a theoretical explanation, as we now clarify in the explanation in p.5, l.121-126.

C: I think that the use of a 2% sample is somewhat critical, since a lot of information is dropped. Why were so many instances dropped, how was this number (2%) chosen, and how can the authors guarantee that this is a representative sample? The full data set should include roughly 2,373,300 records (2% correspond to 47,466 records). A data frame with 2 million rows and maybe 10 columns is definitely still manageable on local machines.

R: The principle reason for using a sample of the OSM buildings for all cities was reducing the time of data processing with GIS to arrive with table of data. Then, the efficiency of a non-parametric Bayesian Network calculation (or a Random Forest computation) drops significantly with large numbers of data points, which would make it less suitable for a possible operational pan-European application. Those aspects notwithstanding, the buildings of 30 capital regions are not the complete population of the European residential buildings, but a small fraction of them, and using all data would still mean using a sample of European buildings. We went back to the full dataset and extracted a larger share of the data, so that the influence of using different dataset size can be presented. The whole population of usable data (i.e. no data missing for any variable at a given location) results in a correlation matrix almost identical correlation matrix, differing by a rank correlation of 0.007 at most. This matrix (N = 2,375,058) is used in revised Table S1.

C: In addition, the 2% sample was only used once. Results were then tested once on a 1% sample. This approach is not very robust. Proper k-fold cross-validation using the full data set would be desirable.

R: We made a new extraction of the data from the full dataset, containing 10% of the data. This sample was used for a 10-fold cross-validation, and the revision of other results. We have revised Tables 2 and 3 (as Tables 3 and 4), and revised section 3.1 accordingly.

C: What was the reason to use a BN for predicting exposure? Was the BN the only model that was tested, or was it contrasted to other approaches? Once the full data set is created, model comparison is comparatively less time-consuming than data preparation. Since Bayesian approaches are often computationally demanding, a classical regression approach or simple machine learning model (e.g. random forest) might be worth trying. This would also allow to investigate more complex interactions between variables as well as non-linear effects.

R: There are several advantages of the class of BNs used here, compared with other approaches: (1) they are probabilistic, providing uncertainty bounds, (2) they can be applied when some of the input data is missing (which Random Forests can't), as e.g. the imperviousness dataset is not gap-free for Europe, which makes it more useful for applications, (3) the whole model can be presented graphically (in Random Forests, only singular trees out of the forest could be presented), (4) accuracy depends only on the configuration of nodes and arcs, and not on model parameters such as number of splits, leaves, trees etc. We applied Random Forests to our data (using Matlab functions 'TreeBagger' and 'predict'), with a 10-fold cross-validation on the basis of the 10% of the data – the same set-up as for the BN. The number of trees was set to 100, maximum number of leaves to 50, in-bag fraction to 1/3, and splits to two. The resulting $R^2$ is slightly lower, RMSE higher, MBE strongly negative and only MAE is slightly lower than when using the BN. Overall, the performance is quite similar, though still worse. The comparative results are now discussed in section 4.1.1 and presented in Table S10.

C: The authors assume that there are no country-specific differences in H, apart from those that are implicitly modelled by including POP, IMD and B. The authors claim that they provide a 'universal method for estimating exposure of residential assets' (p.1, l.3f) across whole Europe. Since the method was only validated with data from Poland, Germany and the Netherlands, I am not sure if this statement is fully justified. Since the characteristics might be different in different countries, using a variable specifying geographical location (e.g. country or even broader geographical region) might be helpful to tackle unobserved heterogeneity.

R: The datasets for Poland, Germany and the Netherlands were the only independent datasets with information on floor space or number of floors for individual buildings that we were able to collect. If the reviewer knows other datasets like those for other countries, we would add them to the analysis. Regarding the heterogeneity, individual models for each country could be made, and they would improve performance at each, but this would still give only a good prediction for the capital city region of a given country, leaving uncertainty how well such a model would perform in other parts of the country. We have moved results for individual cities from the Supplement to the main text (Table 2) to make the variation in the model's quality more visible to the reader. We also note that we discovered and corrected an error related to the dataset German flood-affected households. In the original submission, only those households affected by fluvial floods were included, and not those affected by pluvial floods as suggested by the years provided in revised Table 3. We have added those missing households, increasing the number of records from 2330 to 2868, though the validation results didn't change much.

C: I found the explanation for the empirical relationship given in Eq. (1) a little bit difficult to understand, since the numbers are scattered throughout the paragraph below the formula. I suggest to streamline this explanation.

R: We have rewritten the paragraph to clarify the explanation (p.5, l.136-147).

C: Also, I realized that within Eq. (1), B is used (1.) to derive H, and (2.) to compute F, which is based on H. I don't think that this is a problem, but I noticed that this puts quite a lot of weight on B.

R: The conditional correlation between B and H was considerable, therefore it had to be included in predicting H, and it is indispensable in calculating F, which strongly depends on both B and H.

C: I suggest to include a supplementary table to show which formula for deriving St was used for each country.

R: A supplementary table was added (Table S5).

C: Generally speaking, the coefficient of determination denotes the share of explained variance in the dependent variable that is predictable using independent variable. Note that $R^2 == r^2$ holds only in special cases such as simple linear regression if an intercept is included. While this is the case in the assessment of predicted vs observed values presented in the paper, where the coefficient of determination equals the square of the correlation coefficient, the authors may want to clarify this.

R: The linear regression of revised Fig. 1 is approximately $1.02x - 0.3$ (varies slightly within the 10-fold cross-validation).

C: Being a very common error metric, root mean squared error could be included as well, since it provides more information content with respect to outliers.

R: RMSE was added to the results in Tables 2 and 3.

C: The first two sentences of Section 2.4.2 are unclear to me. The collective out-of-sample validation was done using an unseen 1% sample across all cities. How was the individual validation performed? By using stratified 1% samples of each city? The second sentence starting with 'Then' suggests that the procedure is different and that the samples are not the same. If the same stratified sample is used, validation results can be assessed both city-specific and at an aggregated European level.

R: We replaced the existing results with a 10-fold cross-validation, and corrected the text in the methods and results sections accordingly, and revised Tables 2-4.

C: An overall $R^2$ of 0.36 is moderate, indeed. This means that only a third of the observed variance in building height can be explained using modelled building height (given that observed vs. predicted regression was used). The confusion matrix (Table 3) showing around 25% (and an increasingly lower amount as the number of floors increases) correctly classified outcomes for buildings with more than 2 floors is also slightly puzzling. Again, this might be a hint to try (1.) using more data and (2.) comparing different modelling approaches. Good results for average height are of rather limited explanatory power in terms of model quality assessment, since I would naturally assume that the differences in means are not too large when using any reasonable model. The problem of low variance might also be tackled by (1.) and (2.) mentioned in the previous sentence. That the model does not perform satisfactory at all for cities like Nicosia and Reykjavik might indicate that there are country-specific differences. All cities that exhibit good performance are located in Central Europe (Vienna, Berlin, Amsterdam, Luxembourg, Warsaw, Zagreb).

R: We added more data (10% instead 2%), but this had only marginal effect on the data, as one might expect from a randomized sample of a very large dataset. Several cities have rather poor performance, but it is also partly due to variation in the quality of height data (which routinely shows errors of 1–3 m, according to the validation information contained in the dataset) and OSM buildings (which is particularly poor for Nicosia, for instance, using visual inspection). Some of the cities with good results are not located in Central, but Western and Northern Europe (Amsterdam, Berlin, Luxembourg, Stockholm, Vienna). We revisited the dataset using Random Forests, as described in a previous comment, which didn't lead to improvement. However, we tested the model for different urban-rural typologies using "Degrees of Urbanisation 2014" dataset from Eurostat. This allows classifying our data points according to whether they were located in "cities", "towns and suburbs" or "rural areas" (defined at the level of local administrative units - LAUs). The results are now presented in the supplement (Table

S9) and show that the $R^2$ is lower for towns and rural areas than for cities, though this stems from the much lower variation in building height; MAE is lower in those cases, and in all three types of LAUs MAE has almost the exact same proportion to average height.

C: In the abstract, a validation with (1) buildings in Poland and (2) a sample of Dutch and German houses is mentioned. In the paper, (1) can be found in section 3.3, and (2) is described in the last paragraph of 3.1. I think the title of subsection 3.3 should be reworked, as 'Example application' is rather generic. Maybe a dedicated validation subsection for these new data sources could be helpful?

R: The example in section 3.3 is not based on the validation dataset for Poland. The Polish dataset is from a government-run national database (BDOT) and is used for validation in 3.1 as shown in Tables 2 and 3. The example in 3.3 uses an extraction of OSM data, as the BDOT dataset is not openly available in contrast to OSM. The example is meant to present how the methods from the paper can be used in practice. We now highlight this better in the text.

C: In fact, there does seem to be a slight systematic bias in the results. Figure 2 shows overestimation for low building heights and underestimation of high building heights, with accurate results around 12 m. The regression line likely has a negative intercept and a slope larger than 1.

R: The linear regression of revised Fig. 1 is approximately $1.02x - 0.3$ (varies slightly within the 10-fold cross-validation). We note that the model underestimates the height of tall buildings, but this is partly because few buildings are very tall.

C: The structure of the discussion is generally well thought through. However, the authors again solely focus on the BN model. Maybe the use of other models might lead to better results on the same data set? Limitations of the BN model itself and implications of using a comparatively small sample size (given available data) are not discussed.

R: We added more information about the uncertainty related to the data analysis (section 4.1.1), and we add results generated with Random Forests (section 4.1.1 and Supplementary Table S10).

C: Figure 1: The histogram plots do not have any axis labels and units, which is a major limitation (in terms of information content) of this figure, since the histograms are essentially incomplete. Also - for the sake of consistency: the unit for population density is missing in the caption.

R: The graph and caption were corrected.

C: Figure 2: Please use the same spacing for axis ticks (either steps of 5 or 10).

R: The ticks were corrected.

C: Figure 3: I suggest to use points instead of bars. The information that needs to be transported is the value at the end of the bar, not the area of bar itself. Therefore, information density is higher when using points. Also, the two colors of the bars are different (orange indicating building value in a and yellowish indicating household contents value in b), but the legend matches only the color in b.

R: The legend was corrected and the graphs were reworked, so that points are used instead of bars.

C: Figures 4 & 5: I think it should be mentioned in the caption that values for each country are based on the respective capitals, since this is important when interpreting the results.

R: The values for each are not based on capitals, but on the economic statistics at national level (section 2.2).

C: Figure 7: Legend for a is missing, only legend for b is provided. Again, I suggest to consider using points instead of bars. If points overlap, you may slightly jitter them along the x axis or use some transparency.

R: The legend was corrected and the graphs were reworked, so that points are used instead of bars.

C: Figure 8: It seems that this figure is not referenced in the text. If this is the case (I might have overlooked it), please add a reference in the text. Also, it reveals a substantial difference when compared to the JRC values, this could be explored/discussed further. Again, I suggest to consider using points instead of bars.

R: Fig. 8 (in the revised manuscript, it is Fig. 9) is now correctly mentioned in line 464. The graph was reworked, so that points are used instead of bars. The large difference between JRC and our estimates could be caused by the assumption of a single ratio between building and contents loss (which we show is far from uniform), transposition of this value from an American flood damage model, and possible differences in definition (JRC estimate including more items). This aspect bis now described better (p.14, l. 437-440).

C: Figure 9: It seems that this figure is not referenced in the text. If this is the case (I might have overlooked it), please add a reference in the text.

R: Fig. 9 (in the revised manuscript: Fig. 10) is now correctly mentioned in lines 467 and 574.

C: Table 1: I am wondering why two different sources for 'Population per area' were used. If both are based on the 2011 census, why not using the one with higher resolution if the model is fitted at a building level?

R: the 1 km data are, in most cases, an aggregation of georeferenced records of all enumerated population during the 2011 censuses, therefore represent very accurately the spatial distribution of population. The 100 m dataset is a disaggregated version constructed in the cited study using land cover/use and soil sealing data, therefore introducing modelling error. The 1 km dataset also represents neighbourhood/urban district population, and the 100 m the population of a group of buildings, and the first proved more relevant to represent the dominant type of buildings in an area.

C: Table 3: '% of correctly predicted floors' is confusing. Only the diagonal values indicates the percentage of correctly predicted floors, all other number are simply the percentage of predicted floors?

R: The text in the table (in the revised manuscript: Table 4) was changed to "% of predicted floors within observed floor class"

C: p.7: Lmean should read Lmean, or simply $\bar{L}$.

R: This was corrected to "L_{mean}".

C: Please check consistency regarding capitalization (e.g.: 'Eq.' vs 'eq.'). NHESS manuscript preparation instructions suggest 'Eq.'.

R: We have corrected the capitalization throughout the manuscript.

C: Please format the supplement according to the journal's standards.

R: The supplement was reformatted according to the NHESS Word template.

**Reviewer 2**

C: One of my main concerns is the rather unorganized structure of the manuscript. It deals with different scales, e.g. nationally aggregated data, data from 30 major cities, different validation samples, and a local case study. Moreover, the manuscript has also different time scales. It lists different states of the datasets and includes timeseries of the temporal development of the economic values. This is on the one

side a benefit in terms of the broad scope but hampers the readability for the reader on the other side. I urge the authors to elaborate a more thorough structure of the manuscript to help reader's orientation.

R: We prepared a new graphic with the workflow of the paper (Figure 1), with references to all sections, figures and tables, which is now included at the beginning of the methodology section. We also reorganized sections 2 and 4, so that the two major components of the study are clearly separated.

C: Another concern is the transferability of the model from an urban context to a rural context. This needs to be validated. For instance, Roethlisberger et al (2018) in the same journal state that a difference in the values per square meter between Centre areas (urban areas) and Residential areas of 60%. An alternative is to restrict the title of the manuscript towards urban context of Europe. The conclusion in the abstract "The study shows that the resulting standardized residential exposure values provide much better coverage and consistency compared to previous studies" is not supported by the results.

R: The referenced paper shows a 60% difference in the insured values per square meter of landuse, not of replacement values per usable floor space as in our paper (Table 2 of Roethlisberger et al. 2018). The other authors' findings are not surprising given the higher density of construction in the city/town centers (to which "Centre" class refers) that in other residential areas. Table 3 of Roethlisberger et al. (2018) actually shows lower average insured values per building volume for "Centre" zones (861 CHF per M^3) than in "Residential" areas (897). However, we agree that there are differences between asset values in urban and rural areas, as exemplified by the cited Portuguese data. We have highlighted this aspect better in the discussion (p.18, l.557-560), but at the moment the availability of both regional economic data and local validation data is very limited, and therefore we didn't attempt (yet) to calculate sub-national asset values per m^2.

C:  In the Introduction section, the authors state that the developed procedure is "applicable in any location". This proof is not provided (discussion about urban-rural context). Another main criticism is that the identification of residential buildings is not described. There is an explicit subsection on this topic (section "2.1 Identification of residential buildings"). However, how this identification has been done is described within brackets in section "2.2 Building size estimation" in line 109 ("(identified either through the buildings or the land use layers of OSM)") while in section 2.1 is stated that the identification of buildings and their occupancy (i.e., residential use?) is outside the scope of this paper.

R: We added information on the selection of OSM buildings. Firstly, we downloaded two Map Features ("Buildings" and "Landuse") for the 30 study areas and the example application study from section 3.3. For the analysis, residential buildings were objects from "Buildings" layer which (1) had tags "residential", "apartments", "house", "detached" or "terrace", and (2) had tags "yes" (indicating that a building exists, but the function is not defined) and were located within an object from "Landuse" layer which had a tag "residential". We added this information to p.4, l.90-96.

**Editor**

C: please put particular attention to the overall structure so it will be accessible to the readers of the target journal.

R: We have reorganized sections 2 and 4 and provided Figure 1 as a guide to the different components of the paper.

C: As given in the guidelines of NHESS; I kindly ask you for inclusion of a competing interests statement as follows: "Heidi Kreibich and Kai Schröter are members of the Editorial Board of NHESS" or something similar, please see: https://www.natural-hazards-and-earth-system-sciences.net/about/competing_interests_policy.html (examples section).

R: We added the suggested competing interests statement to the paper.

[revised manuscript text omitted]

---

## Author Response (AR2)

We thank the referee #2 for useful suggestions. We list below the comments (C) and responses (R).

**C:** Equation 1: In the original manuscript, c was 0.7. In the revised version, c=30%. Please double check this value.

**R:** The original manuscript was correct and the revised description was incorrect, as 30% is removed as non-usable space, hence leaving 70% as useful floor space. This was corrected in the text.

**C:** p.11, l.326: I suggest to simply write "low bias" instead of "negligible bias". Negligible might be a little bit too judgemental for a results section. Also, I think that the bias might be negligible from a practical point of view, but from a purely statistical assessment it seems that low building heights are overestimated and tall buildings are underestimated, entailing that a simple linear regression line will be systematically shifted compared to a 1:1 fit (45° line). It might be a good idea to add these two lines to Fig. 3.

**R:** We changed the description of the bias as suggested. We also added the two lines to the graph and amended the caption accordingly.

**C:** p.11, l.328: "An out-of-sample was also carried out ..." - There is a noun missing after sample. Alternatively, I would just refer to this procedure as a standard leave-one-out cross-validation?

**R:** Yes, the word "validation" was missing. We have rewritten the sentence to make it more clear: "An out-of-sample validation was also carried out for each city in the dataset, where the validated capital was left out from the data quantifying the dependency structure of the BN model"

**C:** p. 14, l.33f: Just a thought: Isn't "For comparative purposes" a better motivation than "Due to the moderate accuracy"?

**R:** We agree that this is a more precise explanation. We changed the text as suggested.

**C:** p.14, l.33: I suggest to replace "automated data-mining method" with a term clearer describing random forests, such as e.g. "ensemble learning method". The term "data mining" can be misleading, since it is mainly used in a pattern extraction context.

**R:** We have changed the term with the one suggested by the reviewer.

**C:** p.14. l.36ff: One reason for the low performance of the random forest might be that hyperparameters (such as the number of variables to possibly split at in each node, minimal node size, etc) were not tuned properly, and the rather small number of trees. However, since this is not the main focus of this paper, this is probably not much of an issue.

**R:** We tested different number of trees and more trees had only marginal effect on the results. Nonetheless we agree that more tuning could be done. We added to the text that "with more effort in tuning the various parameters of Random Forests a better result could be achieved.".

**C:** Figure 2: x and y axis labels are still missing.

**R:** We added the axes values in the figures and updated the figure caption.

Also, the acknowledgments were amended to mention the referees.

[revised manuscript text omitted]